# HIV Tat controls RNA Polymerase II and the epigenetic landscape to transcriptionally reprogram target immune cells

Jonathan E Reeder[1,2†], Youn-Tae Kwak[2†‡], Ryan P McNamara[2], Christian V Forst[3], Iván D'Orso[2*]

[1]Department of Biological Sciences, University of Texas at Dallas, Richardson, United States; [2]Department of Microbiology, University of Texas Southwestern Medical Center, Dallas, United States; [3]Department of Genetics and Genomic Sciences, Institute for Genomics and Multiscale Biology, Icahn School of Medicine at Mount Sinai, New York, United States

**Abstract** HIV encodes Tat, a small protein that facilitates viral transcription by binding an RNA structure (*trans*-activating RNA [TAR]) formed on nascent viral pre-messenger RNAs. Besides this well-characterized mechanism, Tat appears to modulate cellular transcription, but the target genes and molecular mechanisms remain poorly understood. We report here that Tat uses unexpected regulatory mechanisms to reprogram target immune cells to promote viral replication and rewire pathways beneficial for the virus. Tat functions through master transcriptional regulators bound at promoters and enhancers, rather than through cellular 'TAR-like' motifs, to both activate and repress gene sets sharing common functional annotations. Despite the complexity of transcriptional regulatory mechanisms in the cell, Tat precisely controls RNA polymerase II recruitment and pause release to fine-tune the initiation and elongation steps in target genes. We propose that a virus with a limited coding capacity has optimized its genome by evolving a small but 'multitasking' protein to simultaneously control viral and cellular transcription.

**\*For correspondence:** Ivan.Dorso@utsouthwestern.edu

[†]These authors contributed equally to this work

**Present address:** [‡]Department of Biochemistry, University of Texas Southwestern Medical Center, Dallas, United States

**Competing interests:** The authors declare that no competing interests exist.

## Introduction

Transcription of protein coding genes by RNA Polymerase (Pol) II is regulated at several steps including initiation, elongation and termination (*Adelman and Lis, 2012*; *Fuda et al., 2009*; *Grunberg and Hahn, 2013*; *Sims et al., 2004*; *Zhou et al., 2012*). Transcription factors coordinate the activation and/or repression of key regulatory programs by acting at one or multiple steps in this regulatory circuitry. While one group of transcription factors recruit Pol II to their target genes to induce transcription initiation, another class functions by promoting the transcriptional pause release from the promoter-proximal state to allow Pol II transition to the productive elongation phase (*Feinberg et al., 1991*; *Gomes et al., 2006*; *Peterlin and Price, 2006*; *Rahl et al., 2010*; *Zhou et al., 2012*). Thus, promoter-proximal pausing has been identified as a general feature of transcription control by Pol II in metazoan cells and a key regulatory step during differentiation, cell development and induction of stem cell pluripotency (*Adelman and Lis, 2012*; *Core and Lis, 2008*; *Smith and Shilatifard, 2013*; *Zeitlinger et al., 2007*; *Zhou et al., 2012*).

In addition to Pol II and the basal transcription machinery, the epigenetic landscape plays another critical role in transcription regulation, mediated by covalent modifications to the N-terminal tails of histones. The study of chromatin modifications has revealed fundamental concepts in the regulation

**eLife digest** The human immunodeficiency virus (HIV) reproduces and spreads throughout the body by hijacking human immune cells and causing them to copy the virus's genetic information. As the virus multiplies, it also causes the death of the immune system cells that help the human body recognize and eliminate viruses. This allows the virus to multiply unchecked.

Studies of the genetic material of HIV – which is in the form of single-stranded RNA molecules and contains only a handful of genes – have begun to reveal how the virus can wreak such havoc to the human immune system. A small protein encoded by the virus, called Tat, boosts the expression of HIV genes in infected immune cells by binding to a structure that forms on newly synthesized viral RNAs. Recent evidence suggests that HIV also changes the expression of human genes to make immune cells more hospitable to the virus. However, it was not known exactly which specific genes are targeted, or how the virus alters their expression.

Now, Reeder, Kwak et al. reveal how the Tat protein alters the expression of more than 400 human genes. Rather than bind to the same structure seen in newly forming HIV RNAs, Tat turns on or off the expression of its human target genes by interacting with proteins that regulate human gene expression. In doing so, Tat is able to precisely control the activity of an enzyme called RNA Polymerase II that is necessary for the early steps of gene expression.

Tat's multitasking ability – boosting HIV gene expression at the same time as reprogramming human gene expression – helps explain how a virus with so little genetic material of its own can perform such a wide range of activities in infected cells. The work of Reeder, Kwak et al. suggests that Tat reshapes the human genome to position target genes in ways that allow them to be efficiently turned on or off. Future studies will further reveal how Tat accomplishes this genome remodeling during different stages of infection. In addition, further research is also necessary to look closely into the sets of genes targeted by Tat to find patterns of genes that work together to alter cell behavior, and investigate how these new behaviors allow HIV to thrive.

of transcription. Histone marks, in general, associate with different genomic domains (promoters, coding units and enhancers) and provide evidence of their transcriptional status (*Barski et al., 2007*; *Creyghton et al., 2010*; *Guenther et al., 2007*; *Li et al., 2007*; *Tessarz and Kouzarides, 2014*; *Zhou et al., 2011*). While active promoters are marked with histone 3 lysine 4 trimethylation (H3K4me3), active transcription units are marked with H3K79me3 and H3K36me3 (*Kouzarides, 2007*; *Li et al., 2007*). Thus, most Pol II regulated genes (protein-coding and long non-coding RNAs) contain a K4/K36 signature that positively correlates with active gene expression (*Guttman and Rinn, 2012*; *Martin and Zhang, 2005*). In addition to promoters, distal genomic elements referred to as enhancers modulate gene activity through gene looping or long-range chromatin interactions and further regulate the location, timing, and levels of gene expression (*Bulger and Groudine, 2011*). The genomic locations of these enhancers (inter- or intra-genic) usually correlate with high H3K4me1 and low H3K4me3 content and their activity is proportional to the levels of H3K27Ac (*Bulger and Groudine, 2011*; *Creyghton et al., 2010*; *Kim et al., 2010b*; *Kowalczyk et al., 2012*; *Smallwood and Ren, 2013*). Additionally, H3K27Ac also appears to modulate two temporally separate events: it enhances the search kinetics of transcriptional activators and accelerates the transition from initiation into elongation leading to a robust and potentially tunable transcriptional response (*Stasevich et al., 2014*).

Viruses have evolved strategies to precisely orchestrate shifts in regulatory programs. In addition to sustaining transcription of their own genomes, certain viral transcription factors directly or indirectly alter existing cellular programs in ways that promote viral processes (replication and spread) (*Ferrari et al., 2008*; *Horwitz et al., 2008*) and/or modulate programs in the infected target cell. It is well established that the HIV Tat protein relieves promoter-proximal pausing at the viral promoter by recruiting the positive transcription elongation factor b (P-TEFb) to the *trans*-activating RNA (TAR) stem-loop formed at the 5'-end of viral nascent pre-mRNAs (*D'Orso and Frankel, 2010*; *Mancebo et al., 1997*; *Ott et al., 2011*; *Zhou et al., 2012*; *Zhu et al., 1997*). Recently, it was discovered that Tat recruits P-TEFb as part of a larger complex referred to as super elongation complex (SEC), which is composed by the MLL-fusion partners involved in leukemia (AF9, AFF4, AFF1, ENL,

and ELL), and PAF1. Although SEC formation relies on P-TEFb, optimal P-TEFb kinase activity towards the Pol II C-terminal domain (CTD) is AF9 dependent, and the MLL-fusion partners and PAF1 are required for Tat transactivation (*He et al., 2010*; *Luo et al., 2012*; *Sobhian et al., 2010*).

Moreover, Tat stimulates transcription complex assembly through recruitment of TATA-binding protein (TBP) in the absence of TBP-associated factors (TAFs) (*Raha et al., 2005*), implying that Tat controls both the initiation and elongation steps of transcription, in agreement with early proposals of Tat increasing transcription initiation, stabilizing elongation and precluding anti-termination (*Kao et al., 1987*; *Laspia et al., 1989*; *Rice and Mathews, 1988*). Thus, Tat has the ability to control multiple stages in the HIV transcriptional cycle to robustly increase transcript synthesis to promote viral replication.

In addition to controlling HIV transcription, Tat appears to modulate cellular gene expression to generate a permissive environment for viral replication and spread, and alter or evade immune responses (*Izmailova et al., 2003*; *Kim et al., 2010a*; *Kim et al., 2013*; *Lopez-Huertas et al., 2010*; *Marban et al., 2011*). One example of Tat-mediated down-regulation is the mannose receptor in macrophages and immature dendritic cells, which plays a key role in host defense against pathogens by mediating their internalization (*Caldwell et al., 2000*). A similar case is Tat's repression of the MHC class I gene promoter, which depends on Tat binding to complexes containing the TBP-associated factor TAFII250/TAF1 (*Weissman et al., 1998*). The interaction of the C-terminal domain of Tat and TAF1 suggests that Tat mediates repression functions through the transcription initiation complex. These, and many other examples, suggest that Tat is capable of altering cellular gene expression via association with factors bound to promoters.

More recently, chip-on-chip and chromatin immunoprecipitation sequencing (ChIP-seq) approaches have revealed that Tat has the ability to bind target genes in the human genome. While these previous studies have provided early glimpses about Tat recruitment to host cell chromatin, a comprehensive description of the direct target genes and the molecular mechanisms remain poorly understood. To address those gaps in knowledge, we aimed at: (i) identifying Tat target genes in the human genome, (ii) delineating the molecular mechanisms by which Tat reprograms cellular transcription, and (iii) defining how Tat is recruited to host cell chromatin. One of the main challenges in the field is to obtain a high-quality, comprehensive ChIP-seq that will provide functional insights. To address this challenge, we analyzed the genome-wide distribution of Tat in the human genome using a technically improved ChIP-seq compared to the previous studies. Moreover, we investigated the molecular mechanisms using a global analysis of chromatin signatures that demarcate the position and activity of genomic domains (enhancers, promoters and coding units), as well as Pol II recruitment and activity (*Barski et al., 2007*; *Creyghton et al., 2010*). We provide, for the first time, evidence that the direct Tat target genes share functional annotations and are regulated by common master transcriptional regulators such as T-cell identity factors. While the Tat stimulated genes (TSG) show a positive role in activating T cells, favoring cell proliferation to promote viral replication and spread, the Tat down-regulated genes (TDG) show critical roles in blunting immune system responses, nucleic acid biogenesis (splicing and translation), and proteasome control. Strikingly, Tat functions as both activator and repressor by modulating Pol II recruitment and/or pause release as well as controlling the activity of chromatin-modifying enzymes to reprogram the epigenetic landscape of the host cell.

Taken together, we propose that a virus with a limited coding capacity has optimized its genome by evolving a small protein (Tat) to perform multiple functions throughout the viral life cycle. Beyond controlling HIV transcription, Tat has evolved unique properties to occupy precise genomic domains (promoters and enhancers) to reprogram cellular transcription using unexpected regulatory mechanisms. We provide the molecular basis of an unprecedented paradigm with critical roles in host-pathogen interactions.

## Results

### Genomic domains occupied and regulated by Tat in CD4+ T cells

Previous studies have proposed that Tat modulates cellular gene expression to generate an environment hospitable for viral replication and spread (*Kim et al., 2010a*, *2013*; *Li et al., 1997*; *Lopez-Huertas et al., 2010*; *Marban et al., 2011*). However, the molecular mechanisms remain poorly

understood. To elucidate how Tat performs these functions we aimed at: (i) identifying genomic domains occupied and regulated by Tat in the human genome, and (ii) defining the mechanisms of transcriptional control. We generated high-quality chromatin immunoprecipitation sequencing (ChIP)-seq datasets in two Jurkat CD4+ T cell lines, inducibly expressing FLAG-tagged Tat or green fluorescent protein (GFP) used as negative control (*Figure 1A*). Importantly, the cell line used expresses low and physiologic levels of Tat, which mimic those detected during HIV infection (*Figure 1—figure supplement 1*). We utilized this minimalistic system in order to more precisely examine the effects of Tat alone rather than in an infection setting. The latter approach may compromise the investigation due to the introduction of other viral products that may also affect cellular behavior and/or alter Tat activity (*Frankel and Young, 1998*).

We established a ChIP-seq pipeline using both a Tat antibody (Ab) previously used in ChIP-seq (*Marban et al., 2011*) and a FLAG Ab. We identified genomic domains enriched with statistically significant occupancy relative to input DNA using the model-based analysis of ChIP-seq (MACS) algorithm with a stringent cutoff (FDR<0.05) (*Zhang et al., 2008*). Unexpectedly, we did not find a significant number of overlapping peaks between the Tat and FLAG ChIP-seq datasets, suggesting that the Tat Ab performs poorly in ChIP, providing a molecular explanation for the low-quality dataset generated by Marban et al. (*Marban et al., 2011*) (*Figure 1—figure supplement 2A–E*). We thus focused on FLAG, which enabled us to generate a better quality, more comprehensive ChIP-seq dataset compared with previous studies (*Kim et al., 2010a*; *Marban et al., 2011*), which served as the key basis to elucidate novel Tat functions in the control of cellular transcription. To precisely map Tat binding sites in the human genome, we called FLAG peaks in both the Tat and GFP cell lines, then discarded those peaks in the intersection of both datasets, considering only the FLAG peaks unique to the Tat cell line to be true Tat binding events. We found that the majority of FLAG peaks in the Tat cell line represent bona fide Tat binding sites because they are not detected in the GFP cell line (*Figure 1—figure supplement 2A,C–E*). Importantly, the ChIP signal for FLAG was barely detectable in the GFP cell line or in cells expressing inactivating Tat mutations in the activation domain (such as C22A) that abolish Tat recruitment to target genes and/or impair gene expression activation (*D'Orso et al., 2012*) (*Figure 1—figure supplement 3*). In contrast to the C22A non-functional Tat mutant, synonymous mutations within the RNA-binding domain (such as K50Q and R52R53K) have less pronounced effects on Tat binding to chromatin (*Figure 1—figure supplement 3*).

Genome-wide distribution analysis based on the ENCODE annotation (*Consortium, 2011*), revealed that Tat binding is heavily enriched at promoters (24%, p-value $3.4 \times 10^{-324}$) and 5'-UTR (1.2%, p-value $5.3 \times 10^{-193}$) relative to their genomic proportions. These genomic domains are underrepresented in the genome (about 1%) making Tat's relative abundance at these sites highly significant (*Figure 1B*). Although Tat also binds introns (36.4%, p-value $5.1 \times 10^{-4}$) and intergenic domains (35%, below the expected level of enrichment), defined as sites located more than 1 Kb from the nearest annotated gene, these two genomic domains are highly represented in the genome and the relative Tat binding frequency here is not nearly as striking as at promoters and promoter-proximal regions (*Figure 1B* and *Table 1*).

To gain insights into the roles of Tat in cellular transcriptional control, we integrated the FLAG ChIP-seq dataset with a whole transcriptome generated by RNA-seq (*Figure 1C*). RNA-seq revealed 2013 differentially expressed genes (DEGs) using a q-value cutoff <0.05, 456 of which also appeared in the set of Tat-bound genes generated by our ChIP-seq analysis. We refer to this dataset as direct targets. Remarkably, inactivating mutations that abolish Tat recruitment to host cell chromatin also impair gene expression changes, indicating that the effect of Tat on target gene transcription is direct and requires Tat binding and activity. Specifically, reduced Tat C22A binding to promoters of Tat target genes correlates with decreased gene expression changes (*Figure 1—figure supplement 3*), thus providing direct evidence of Tat function.

Besides the direct target genes, we identified another set referred to as indirect targets. These are genes that are differentially expressed in the presence of Tat (RNA-seq), but are not directly bound by Tat (*Figure 1C*). Tat might regulate these genes through downstream effects or alternative mechanisms (i.e. signaling pathways) or they could be targets not identified by ChIP-seq because of the high-confidence threshold used during peak calling. In this work, we only focused on the direct target genes to study the mechanisms of transcriptional control by Tat.

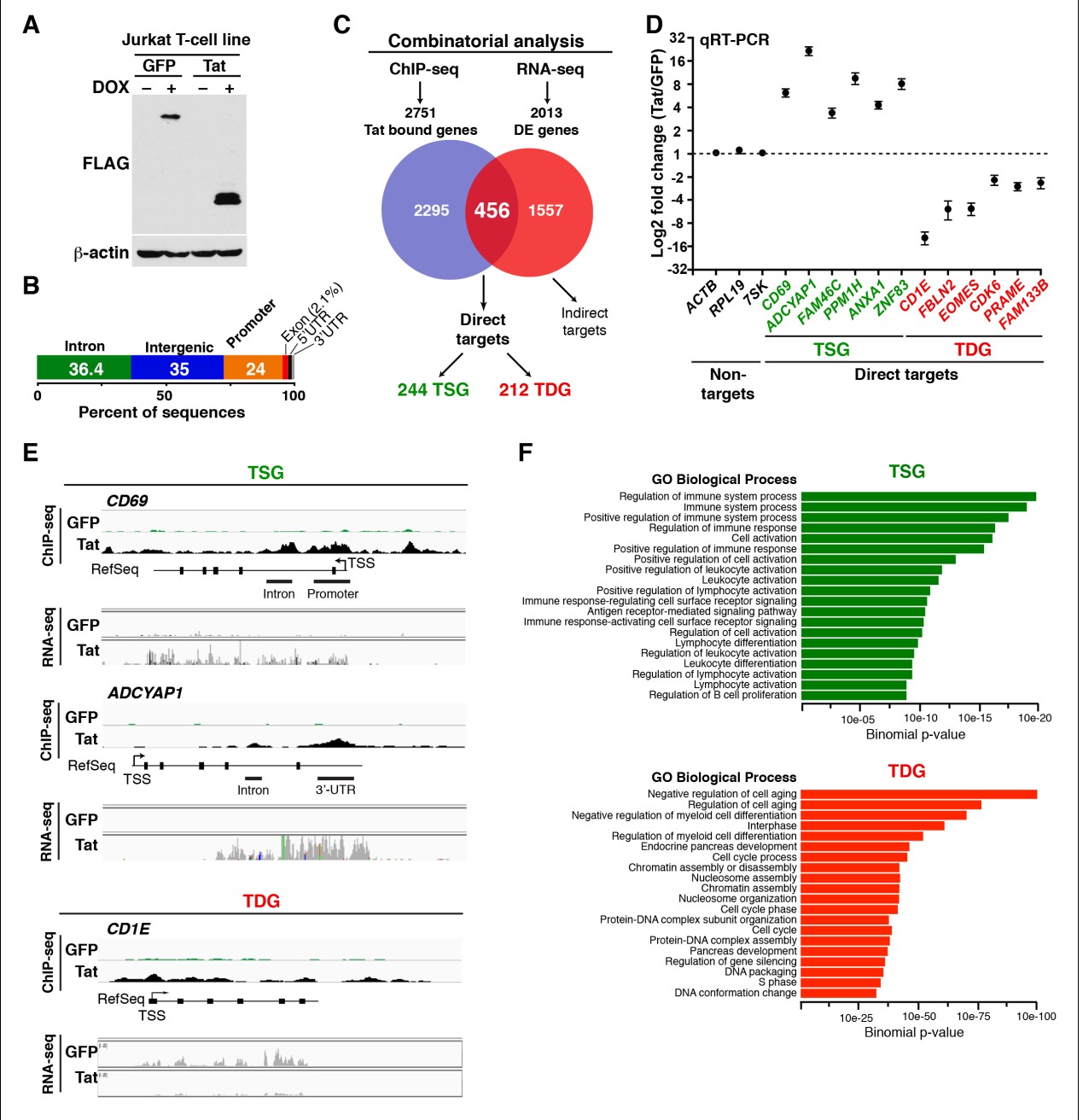

**Figure 1.** Genomic domains occupied and regulated by Tat in CD4+ T cells. (**A**) Western blot of Jurkat-GFP and -Tat cell lines treated (+) or not (–) with DOX using the indicated antibodies. (**B**) Genome-wide distribution of Tat across the human genome. (**C**) Integration of the FLAG ChIP-seq and RNA-seq datasets defines a set of genes directly regulated by Tat. (**D**) Validation of the RNA-seq dataset using qRT-PCR on the indicated TSG, TDG or non-target genes as negative controls (mean ± SEM; n = 3). (**E**) Individual tracks showing FLAG ChIP-seq and the corresponding RNA-seq dataset in the GFP and Tat cell lines. (**F**) Functional annotation of biological processes enriched at TSG and TDG. This figure is associated with *Figure 1—figure supplements 1–10*. Direct targets, genes directly bound and regulated by Tat; ChIP-seq, chromatin immunoprecipitation sequencing; DOX, doxycycline; GFP, green fluorescent protein; RNA-seq, RNA sequencing; qRT-PCR, quantitative real time polymerase chain reaction; TDG, Tat downregulated genes; TSG, Tat stimulated genes; TSS, transcription start site.

The following figure supplements are available for figure 1:

**Figure supplement 1.** Tat protein expression levels in the Jurkat Tat-SF model matches the levels of Tat detected during HIV infection.

**Figure supplement 2.** Technical improvement of Tat ChIP-seq in CD4+ T cells.

*Figure 1 continued on next page*

*Figure 1 continued*

**Figure supplement 3.** Non-functional Tat mutants have compromised chromatin interaction and modulation of cellular gene expression.

**Figure supplement 4.** Tat-induced transcriptome changes are also observed at the protein level.

**Figure supplement 5.** Distribution of Tat occupancy at promoter and/or intragenic domains in TSG and TDG.

**Figure supplement 6.** FLAG ChIP-qPCR analysis on the indicated genomic loci.

**Figure supplement 7.** The genes modulated by ectopic expression of Tat are also detected during a time-course HIV infection experiment.

**Figure supplement 8.** HIV infection of central memory CD4+ T cells triggers deregulation of TSG and TDG detected in the genome-wide approaches.

**Figure supplement 9.** Response network of TSG and TDG.

**Figure supplement 10.** Enrichment of TSG and TDG in publicly available datasets of differentially expressed genes identified in HIV infection and replication experiments.

Further analysis of the direct Tat targets revealed that 244 genes are up-regulated and 212 genes down-regulated, and we refer to them as Tat stimulated genes (TSG) and Tat downregulated genes (TDG), respectively (*Figure 1C*). Importantly, we validated the expression of several direct target genes using quantitative real-time polymerase chain reaction (qRT-PCR) assays, confirming the reliability of RNA-seq (*Figure 1D*). Notably, protein expression analysis revealed that the changes detected at the RNA level are also reflected at the protein level (for example CD69 expression at the cell surface in the presence of Tat) indicating that gene expression changes are functional (*Figure 1—figure supplement 4*).

We further analyzed the distribution of Tat binding sites within the direct target genes to probe for enrichment in particular genomic domains. We found that Tat is equally recruited to both promoters (39%) and intragenic domains (39%) at TSG, whereas the majority of occupied domains at TDG are at promoters (61%) (*Figure 1—figure supplement 5*). Inspection of genome browser tracks showed that, in addition to binding TSG promoters (*CD69*), Tat is also recruited to gene body regions with enrichment at introns (*ADCYAP1*) (*Figure 1E*). Notably, this mode of Tat binding appears to be functionally relevant because it correlates with RNA abundance changes as revealed by RNA-seq. In addition, Tat binds the target genes using discrete (*CD69* and *ADCYAP1*) or broad (*CD1E*) distribution patterns, probably due to the different modes or mechanisms of recruitment to chromatin (*Figure 1E*). Importantly, we have validated the FLAG ChIP-seq dataset by performing

**Table 1.** Genome-wide distribution of Tat in CD4+ T cells.

| Genomic domain | | Peaks | Percent | Genome fraction | ChIP (p-value) |
|---|---|---|---|---|---|
| Total | | 6117 | 100 | | |
| Intergenic | | 2141 | 35 | 52.7% | |
| Intragenic | Promoter (−1 kb–TSS) | 1469 | 24 | 1.1% | $3.4 \times 10^{-323}$ |
| | Exon | 128 | 2.1 | 1.9% | $1.4 \times 10^{-3}$ |
| | Intron | 2227 | 36.4 | 42.4% | $5.1 \times 10^{-4}$ |
| | 5′ UTR | 73 | 1.2 | 0.4% | $5.3 \times 10^{-193}$ |
| | 3′ UTR | 79 | 1.3 | 1.5% | $1.4 \times 10^{-2}$ |

ChIP-seq, chromatin immunoprecipitation sequencing; TSS, transcription start site; UTR, untranslated region.

extensive ChIP-qPCR on several direct targets including two TSG (*CD69* and *ADCYAP1*) and two TDG (*CD1E* and *RAG1*) in the Tat and GFP cell lines (*Figure 1—figure supplement 6*).

Given that our model was built on the ectopic expression of Tat in target cells, we performed an infection experiment to test whether Tat reprograms cellular transcription in a similar way in the context of infection. Importantly, the TSG (*CD69, FAM46C and PPM1H*) and TDG (*CD1E, EOMES and FBLN2*) tested are also modulated early during HIV infection of Jurkat T cells, albeit with different kinetics (*Figure 1—figure supplement 7*), supporting the view that the cellular reprogramming by Tat is functional and not simply an artifact of RNA-seq or the ectopic expression of Tat outside the context of the virus.

Given that we proposed that Tat effects on host cell gene expression might be relevant to normal HIV biology we asked whether the TSG and TDG are also modulated in response to viral infection of primary CD4+ T cells. To this end, we isolated naïve CD4+ T cells from the blood of healthy donors and generated central memory T cells ($T_{CM}$) (*Figure 1—figure supplement 8A*). After infection of $T_{CM}$ with replication competent X4 trophic virus (NL-GFP) or mock infection and sorting infected cells, we isolated RNA and performed qRT-PCR analysis on several TSG and TDG. We observed that HIV infected $T_{CM}$ cells showed the differential gene expression signature (at least for the 12 direct targets examined) that we previously observed with the ectopic expression of Tat or HIV infection of Jurkat T cells (*Figure 1—figure supplement 8B*). This *ex vivo* experiment clearly demonstrates the robustness of our minimalistic setting to study Tat functions in the host cell.

## The direct Tat target genes share common functional annotations and are enriched in pathways beneficial for the virus

If the genes directly modulated by Tat are involved in biologically relevant processes then we would expect them to share functional annotations. To explore whether the TSG and TDG have any common biological functions we examined their gene ontology (*Figure 1F*). To provide statistical robustness, we used cluster analysis and a control set of genes depleted in the Tat ChIP-seq experiment. Gene categories significantly enriched in the set of TSG include positive regulation of immune system process, cell activation and regulation of lymphocyte differentiation, while TDG include negative regulation of cell aging, regulation of myeloid cell differentiation and processes of DNA/RNA biogenesis (*Figure 1F*). Consistently, network analysis indicates that TSG are significantly enriched in T-cell receptor (TCR) pathway, cell cycle and focal adhesion, while TDG enrich processes relevant for DNA/RNA processes, ribosome and proteasome control, among others (*Figure 1—figure supplement 9*).

With respect to T-cell activation, CD69 exhibits a rather central role, because its upregulation promotes T-cell stimulation and differentiation (TCR pathway cluster) (*Sancho et al., 2005*). Another stimulated process involves components of the cell cycle (CDK6) together with cyclinD3 (CCND3) and cyclin-dependent kinase inhibitor 1B (CDKN1B) (cell cycle cluster) that appear to be controlled by phosphorylation via the lymphocyte-specific protein tyrosine kinase (LCK) from the TCR complex, as one of the central node in the network. Another controller node assembles the ataxia-telangiectasia-mutated (ATM) serine/threonine kinase, which is best known for its role as an activator of the DNA damage response (HIV infection cluster). The activity of HIV integrase stimulates an ATM-dependent DNA damage response, and ATM deficiency sensitizes cells to retrovirus-induced cell death. In addition, ATM inhibition is capable of suppressing the replication of both wild-type and drug-resistant HIV (*Lau et al., 2005*), thus demonstrating the importance of this TSG in controlling viral processes. With respect to down-regulated processes, ribosomal proteins centered around RPS9 (ribosome cluster), together with translation initiation factors (EIF3b) and the nucleolar and coiled-body phosphoprotein NOLC1, as well as components of the spliceosome such as SF3B5, SNRPB, SNRNP200, LSM4, and PCBP2 (spliceosome cluster), suggest negative regulation of these processes (*Figure 1—figure supplement 9*).

Because we proposed that the predicted GO biological processes of the direct Tat target genes are essential for the viral life cycle (to promote a permissive state for viral replication), we further analyzed whether they are retained in the context of infection. To test this, we assembled a collection of 62 publicly available datasets including 48 gene-sets from 13 publications containing information on DEGs identified during HIV infection together with 14 datasets from the Molecular Signatures Database (MSigDB) of the Broad Institute (*Subramanian et al., 2005*) (*Supplementary file 1A*). After executing gene-set enrichment analysis with Bonferroni correction

for multiple testing, we identified five datasets that significantly enriched the TSG (with FDR ≤ 0.05) (*Supplementary file 1B*). In addition, TDG also enrich two gene sets from the HIV relevant MSigDB sets (*Figure 1—figure supplement 10A* and *Supplementary file 1B*). Together, the analysis provides evidence that the direct Tat target genes (and thus the predicted GO biological processes) are retained in the context of viral infection, supporting the model that Tat-mediated host cell reprogramming occurs during infection. It is noteworthy this proposal is also consistent with our infection data on Jurkat and primary $T_{CM}$ cells (*Figure 1—figure supplements 7,8*).

Our network analysis indicated that TSG and TDG were enriched in specific biological processes that might promote viral functions (including replication) (*Figure 1—figure supplements 9,10*). To answer how those functional annotations could promote viral infection, we employed a variety of methods to identify the functions of the direct Tat target genes, and relate them to the biology of HIV. Functional annotation by GO classes provides an overview of biological processes and functions enriched by the gene-sets (*Figure 1—figure supplement 10B*). Furthermore, we have obtained additional information from the 'canonical pathway' collection of MSigDB. By annotating clusters of the response network with GO and MSigDB pathways we related biological processes with network content. For example, the TCR pathway cluster annotated with MSigDB includes ITK, LCK, LCP2, PRKCQ and VAV3, among other targets. This cluster not only reveals that those targets interact with each other, but also with the Tyrosine kinase LCK, which is known to phosphorylate both LCP2 and PRKCQ, implying physical and functional interactions.

The data clearly indicates that TSG are enriched in datasets from many pathways that correlate with stimulation of viral replication, such as the 'TCR pathway/stimulation' (CD3, CD247, INPP5D, LCP2, and PTEN) and 'downstream TCR signaling/pathway' (PIK3CA and PRKCQ), 'T-cell co-stimulation' (CTLA4 and CD28), 'cell motility signaling/pathway' (RAC1), 'generation of second messengers' (EVL, ITK, and LCK), and 'phosphatidylinositol signaling' (INPP4A, INPP5D, PIK3CA, PLCB1, PRKCB, and PTEN), among others (*Figure 1—figure supplement 10B* and *Supplementary file 1B*). The prime example of pathways enriched in TSG is T-cell signaling/activation and co-stimulatory signals (CD28) that provide additional control mechanisms to prevent inappropriate and hazardous T-cell activation. The data is consistent with the fact that, in early stages of infection, the viral encoded proteins (particularly Tat) mimic T-cell signaling pathways, resulting in sustained viral replication within infected T cells. This T-cell activation provides new targets for HIV replication, creating a favorable environment for further virus-mediated damage to the immune system and chronic consumption of the pools of naïve and resting memory cells (*Hazenberg et al., 2000*; *Hellerstein et al., 2003*). Our data suggest that HIV uses Tat to directly induce several pathways (including T-cell signaling) for productive infection of immune cells. Both the biological data and our prediction suggest that TSG play important roles in these processes. On the other hand, TDG are primarily enriched in datasets related to 'metabolism of RNA' (HSPA8, LSM4, and PRMT5), in particular 'ribosome' (RPL and RPS variants), 'proteasome' (PSMB3, PSMB4, PSMB5, PSMC2, PSMD1, and PSMD8), and 'regulation of apoptosis' (DAPK1, together with above proteasomal genes), among others identified by the HIV life cycle/host interaction gene sets from Reactome (*Croft et al., 2014*) (*Figure 1—figure supplement 10B* and *Supplementary file 1C,D*). However, further studies will be required to precisely define the role of individual TSG and TDG on viral replication.

Taken together, we have established the first comprehensive framework of target genes in the human genome directly bound and regulated by Tat. Below we study the molecular mechanisms of transcriptional control by examining genome-wide changes of Pol II and chromatin signatures.

## Global analysis of chromatin signatures indicates that Tat stimulates transcription at both the initiation and elongation steps

With these functional insights we focused on direct target genes to delineate the mechanisms by which Tat regulates cellular transcription. Profiling of histone modifications has revealed fundamental concepts in the regulation of transcription (*Barski et al., 2007*; *Zhou et al., 2011*). To investigate whether Tat acts at the initiation or elongation steps, we performed ChIP-seq of several histone modifications (in the absence and presence of Tat) that demarcate distinct genomic domains such as promoters, coding units and enhancers and provide (in general) information regarding their transcriptional status (*Table 2*).

Active promoters usually have adjacent nucleosomes bearing an H3K4me3 transcription initiation mark (*Bernstein et al., 2002*; *Guenther et al., 2007*; *Mikkelsen et al., 2007*; *Santos-Rosa et al.,*

**Table 2.** Genome-wide distribution and function of histone modifications.

| Histone modification | Location | Function |
| --- | --- | --- |
| H3K4me3 | Promoters | Transcription activation |
| H3K79me3 | Gene bodies | Transcription activation |
| H3K36me3 | Gene bodies | Transcription activation |
| H3K4me1 | Enhancers | Does not demarcate active status, but location |
| H3K27Ac | Enhancers | Transcription activation |
| H3K9me3 | Ubiquitous | Transcription repression |

*2002*), and genes marked by H3K4me3 display significant amounts of transcriptionally competent Pol II at promoters (*Min et al., 2011*). Thus, we first determined the levels of H3K4me3 in both cell lines to test whether Tat modulates transcription initiation (*Figure 2A*). After fixing TSS selection for a subset of TSG such as *ADCYAP1* and *ARHGEF7*, where transcription starts at a short, internal iso-form (*Figure 2—figure supplement 1A,B*), we observed that the majority of TSG (excluding *ATP9A, CD244, ADCYAP1, CD226, SERINC2*) are already marked with variable (low-to-high) H3K4me3 levels and, as expected, its distribution mirrors the location of the promoter-adjacent nucleosomes (*Figure 2B*). We thus defined two groups of genes based on H3K4me3 fold change levels in the presence and absence of Tat (Tat/GFP). While the first group of genes (n = 17), referred to as class I TSG, has undetectable or low H3K4me3 levels in the absence of Tat (*Figure 2A* and *Figure 2—figure supplement 1C,D*), the second group of genes, referred to as class II TSG, shows medium-to-high H3K4me3 levels surrounding the TSS (*Figure 2A,B*).

In the presence of Tat, class I TSG experiences a large increase in H3K4me3 density at promoter-proximal regions (2-to 10-fold change) (*Figure 2A,B*, and *Figure 2—figure supplement 1C,D*). Interestingly, Tat can selectively promote *de novo* transcription initiation at a small subset of genes (such as *ADCYAP1, ATP9A* and *CD244*), as revealed by the large fold changes in H3K4me3 and selection of a non-canonical or novel TSS (*Figure 2—figure supplement 1A,C,D*). For example, Tat induces H3K4me3 at an internal site (intron 4) of *ADCYAP1* but not at the canonical TSS, which coincides with the production of a novel, short isoform of a yet unknown function (*Figure 2—figure supplement 1A*). In contrast to class I TSG, class II shows no change or, unexpectedly, a slight reduction in H3K4me3 levels in the presence of Tat (~1.5–2–fold decrease) (*Figure 2A,B*), suggesting a post-initiation role for Tat in activating these genes (see below).

The H3K4me3 density surrounding the TSS of class I TSG was typically lower (at least 5-fold less) than the signal in actively transcribed genes. As expected, these genes are transcriptionally inactive or show low RNA levels (<10 Fragments Per Kilobase of transcript per Million mapped reads [FPKM]) such as signaling peptide hormones (*ADCYAP1*), transcription factors involved in immune system maturation (*ZNF521, RORβ, ETV6*) and genes essential for T-cell maturation and responses (*CD69, CD244*). Interestingly, we observed a positive correlation ($R^2$ = 0.764) between the increase in transcript levels and H3K4me3 density nearby the promoters of these genes, and that highly stimulated genes (*ADCYAP1* and *CD69*) experience larger increase in H3K4me3 density compared with other genes (*Figure 2—figure supplement 2A*). On the other hand, class II TSG shows a much lower correlation ($R^2$ = 0.177) between the increase in transcript levels and H3K4me3 density surrounding promoters (*Figure 2—figure supplement 2B*), consistent with the idea that they are regulated at a post-initiation step (see below).

If increased transcription initiation at these genes correlates with a productive increase in RNA levels, then we would expect to find nucleosome modifications associated with transcription elongation throughout the coding units (*Li et al., 2007*). Previous studies have elucidated that in actively transcribed genes, H3K79me2/3-modified nucleosomes are present at their highest levels shortly downstream of the TSS and that H3K36me3 modifications occupy the entire gene body, with increasing density towards the 3'-end of the gene (*Guenther et al., 2007*; *Kolasinska-Zwierz et al., 2009*; *Li et al., 2007*; *Seila et al., 2008*; *Zhou et al., 2011*). To examine transcription states in detail, we carried out ChIP-seq experiments of H3K79me3 and H3K36me3 using validated antibodies (*Egelhofer et al., 2011*). To assess Tat's role in coupling transcription initiation with elongation at

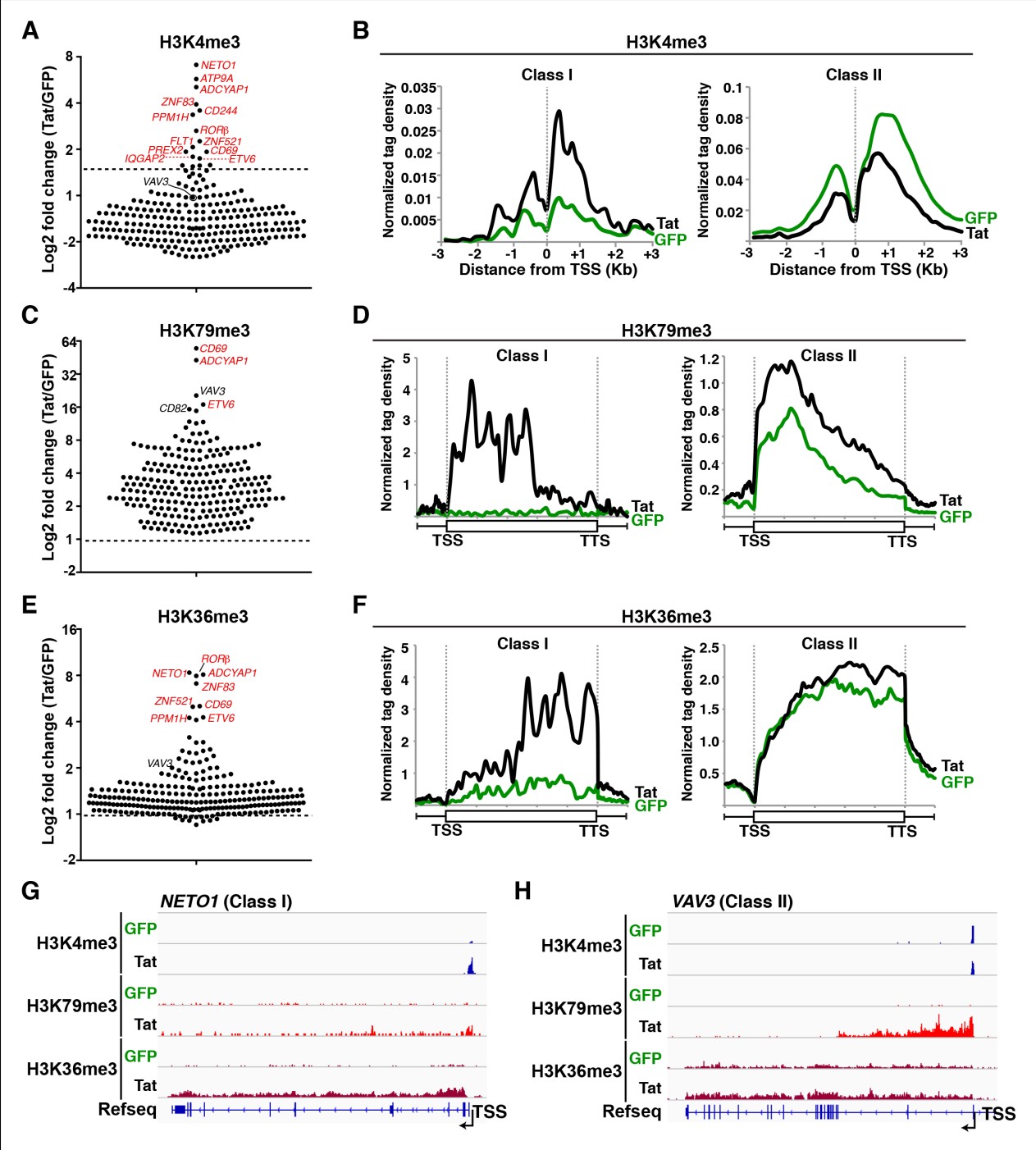

**Figure 2.** Global analysis of chromatin signatures reveals that Tat activates the transcription initiation and elongation steps. (**A**) Dot plots of H3K4me3 log2 fold change in the region encompassing ±3 Kb from the TSS of all TSG. Genes are divided into two clusters: class I and II based on increased (>1.5-fold change) or no change/decreased H3K4me3 levels in the presence of Tat. Selected TSG examples are indicated in red. (**B**) Metagene plots centered on TSS showing H3K4me3 occupancy profiles at both class I (n = 17) and class II (n = 43) TSG in the presence of Tat or GFP. (**C**) Dot plots of H3K79me3 log2 fold change in the region from TSS to TTS (see Materials and methods). (**D**) Metagene analysis showing average H3K79me3 ChIP-seq signals at both class I and II TSG in the presence of Tat or GFP. Units are mean tags per million ChIP-seq reads per bin across the transcribed region of each gene with 2 kb upstream and downstream flanking regions. (**E**) Dot plots of H3K36me3 log2 fold change in the region from TSS to TTS. (**F**) Metagene plots showing average H3K36me3 ChIP-seq signals at both class I and cII TSG in the presence of Tat or GFP. (**G**) Genome browser views showing ChIP-seq signal at a class I TSG (*NETO1*) in the GFP and Tat cell lines. (**H**) Genome browser views showing ChIP-seq signal at a class II TSG (*VAV3*) in the GFP and Tat cell lines. This figure is associated with *Figure 2—figure supplements 1–5*. ChIP-seq, chromatin immunoprecipitation sequencing; GFP, green fluorescent protein; TSG, Tat stimulated genes; TSS, transcription start site; TTS, transcription termination site.

*Figure 2 continued on next page*

*Figure 2 continued*

The following figure supplements are available for figure 2:

**Figure supplement 1.** Tat specifies TSS selection and synthesis of alternate isoforms.

**Figure supplement 2.** Correlation between gene expression levels and H3K4me3 density surrounding the TSS at class I and II TSG.

**Figure supplement 3.** Evidence that Tat increases Pol II and P-TEFb recruitment, and chromatin marks coinciding with transcription initiation and elongation at class I TSG.

**Figure supplement 4.** Transcription initiation correlates with increased elongation chromatin markers.

**Figure supplement 5.** Tat recruits chromatin-modifying enzymes and elongation factors at selected target genes to promote transcription elongation.

class I TSG, we calculated the fold change in H3K79me3 and H3K36me3 tag density in the presence and absence of Tat (Tat/GFP) as well as changes in their gene average distribution in both class I and II TSG (*Figure 2C–F*). As expected, we observed that class I TSG (activated at the initiation step) also shows evidence of transcription elongation, based on the increase in H3K79me3/H3K36me3, total Pol II and elongating Pol II levels, as well as recruitment of the P-TEFb kinase at two TSG (*CD69* and *ADCYAP1*) (*Figure 2—figure supplement 3*). While in the absence of Tat, class I TSG are devoid of H3K79me3 throughout the gene, Tat promotes a robust increase in H3K79me3 (~6.2-fold over GFP) just downstream of the TSS with a progressive decline towards the transcription termination site (TTS) (*Figure 2C–F*, and *Figure 2—figure supplement 4*). Similarly, levels of H3K36me3 at class I TSG are low in the gene body and increase towards the 3'-end of the gene in the presence of Tat (~5.3-fold over GFP) (*Figure 2F*). Remarkably, the average distribution patterns of both H3K79me3 and H3K36me3 in these genes are consistent with previous genome-wide distribution analysis (*Kouzarides, 2007*; *Li et al., 2007*).

Because H3K4me3 levels do not increase in the class II TSG, we reasoned that these genes are regulated by Tat at a post-initiation step. If this were the case, then we would expect to find an increase in nucleosome modifications associated with transcription elongation marks in the presence of Tat, despite the lack of H3K4me3 increase. To test this possibility, we examined H3K36me3 and H3K79me3 distribution and density in these genes. We ignored genes that lack H3K79me3 irrespective of Tat presence as well as intronless genes, which complicated the density and distribution calculations, and thus ended with a more cohesive and consistently behaved group of class II TSG (n = 43). As expected, most class II TSG showed increased H3K79me3 (100%) and H3K36me3 (~82%) levels within their gene bodies (*Figure 2C–F*) consistent with a role of Tat in promoting transcription elongation. Together, this analysis suggests that the majority of target genes that do not experience transcription initiation changes in response to Tat do show evidence of increased levels of promoter escape and transition to active elongation based on the global analysis of chromatin signatures (*Figure 2—figure supplement 4*) and Pol II (see below). Although the increase in H3K79me3 is significant, H3K36me3 changes are smaller, probably because class II TSG is active and their transcribing units already demarcated by H3K36me3 (*Figure 2C–F*). Again, the average gene distribution of H3K79me3 and H3K36me3 correlates with previous studies suggesting that H3K79me3 is most enriched shortly downstream of the TSS and H3K36me3 increases towards the middle and end of the gene (*Figure 2D–F*) (*Kouzarides, 2007*; *Li et al., 2007*).

Inspection of individual gene tracks from our genome-wide study reveals examples of these two regulatory mechanisms (*Figure 2G,H*). At the neuropilin and tolloid-like 1 (*NETO1*) locus (class I TSG), which encodes a transmembrane receptor that plays a critical role in spatial learning and memory, Tat binding at the promoter-proximal region correlates with a sharp increase in H3K4me3 (~8-fold) with concomitant increases in H3K79me3 and H3K36me3 downstream of the TSS (*Figure 2G*). Conversely, at the *VAV3* guanine nucleotide exchange factor locus (class II TSG), the H3K4me3 signature surrounding the TSS does not appear to change, while the markers of active transcription in coding units (H3K79me3 and H3K36me3) showed large increases in the presence of Tat, indicating a role in transcription elongation (*Figure 2H*). Heatmap representation of the density of all three

chromatin signatures at promoter and gene body of class I and class II TSG demonstrate that this is common for all target genes, albeit with differences in fold change (*Figure 2—figure supplement 4A–E*), which is in agreement with the broad distribution of gene expression changes (*Figure 1D*).

Given that we defined two classes of genes using the minimalistic Tat ectopic expression system, we asked whether these genes are also modified in a similar manner (at the initiation or elongation steps) during HIV infection. To test this, we analyzed the levels of initiating (promoter-proximal) and elongating (promoter-distal) transcripts in primary $T_{CM}$ cells infected with replication-competent HIV versus mock infection. Interestingly, we observed that two class I TSG (*CD69* and *PPM1H*) and class II TSG (*VAV3* and *ANXA1*) showed evidence of increased initiation or elongation, respectively (*Figure 1—figure supplement 8C,D*). This implies that the model of cellular reprogramming by the ectopic expression of Tat alone is mirrored (at least for the target genes examined) during HIV infection of primary T cells.

Several enzymes are known to regulate histone modifications associated with distinct epigenetic states. If Tat promotes cellular reprogramming by modifying the epigenetic landscape then we would expect Tat to interact with chromatin-modifying enzymes associated with the respective histone modifications. While H3K79me3 is generated by the Dot1L complex (*Nguyen and Zhang, 2011*), the H3K36me3 mark is imposed by the SetD2 methyltransferase recruited by elongating Pol II and enriched within the body of transcriptionally active genes (*Guenther et al., 2007*; *Krogan et al., 2003a*; *Krogan et al., 2003b*; *Nguyen and Zhang, 2011*; *Pokholok et al., 2005*). To test if Tat does indeed interact with these enzymes, we first affinity purified (AP) Strep-tagged Tat (or GFP, used as negative control) from nuclear fractions of the Jurkat T-cell lines and observed that Tat, but not GFP, binds both Dot1L and SetD2 as well as the P-TEFb kinase (CDK9) used as positive control in this assay (*Figure 2—figure supplement 5A*). To further test whether changes in the epigenetic landscape directly correlate with the recruitment of these chromatin modifiers to specific genes, we performed ChIP followed by quantitative PCR (ChIP-qPCR) and found that the Tat-mediated recruitment of Dot1L and SetD2 to the *CD69* locus (class I TSG) correlates well with the increase in the histone modifications associated with transcription elongation as well as Pol II (*Figure 2—figure supplement 5B,C*). To provide a functional link between the Tat-mediated recruitment of these chromatin modifiers and gene expression changes, we used short hairpin RNAs (shRNAs) to target Dot1L and SetD2 by RNA interference (*Figure 2—figure supplement 5D–H*). Interestingly, we noted that after knockdown of these chromatin-modifying enzymes as well as of CDK9, the increase in CD69 RNA levels (but not RPL19) in response to Tat is virtually abolished, implying that Tat-mediated recruitment of Dot1L, SetD2 and P-TEFb to TSG is a requisite for their increased transcription elongation levels, which is in agreement with the ChIP data (*Figure 2—figure supplement 3*).

The discovery of these key enzymes as Tat targets and the evidence that they play important roles in Tat-mediated cellular gene expression alterations suggest that this regulatory mechanism is part of the reprogramming of target immune cells by Tat. Collectively, we have described a set of human genes stimulated by Tat at two different steps in the transcription cycle (initiation and elongation) using selective chromatin-modifying enzymes and the transcription elongation machinery.

## Tat promotes Pol II recruitment and pause release to induce initiation and elongation at different gene classes

Pol II regulates the control of transcription initiation and elongation in the context of chromatin (*Fuda et al., 2009*). In fact, the deposition of histone modifications associated with initiation and elongation at promoters and gene bodies has been functionally linked to levels of recruited and transcriptionally engaged Pol II, respectively (*Adelman and Lis, 2012*). Therefore, if Tat promotes transcription initiation and elongation as proposed (*Figure 2*), we would expect Pol II levels to fluctuate in response to Tat in ways that reflect the specific mode of activation. Pol II is recruited to gene promoters to initiate transcription, but also tends to occupy promoter-proximal regions of genes that show evidence of initiation but no or inefficient elongation, which is referred to as promoter-proximal pausing (*Adelman and Lis, 2012*; *Fuda et al., 2009*; *Rahl et al., 2010*; *Wade and Struhl, 2008*). To further investigate whether Tat promotes Pol II recruitment (indicative of transcription initiation) or increases Pol II levels within gene bodies (indicative of transcription elongation) we used ChIP-seq to examine Pol II distribution in the absence and presence of Tat (*Figure 3*).

We first analyzed levels of Pol II at both Tat binding sites (Tat peak) and promoters of target genes in cases where Tat only binds to promoter-distal sites. Interestingly, we found that Tat stimulates Pol II recruitment at both the Tat binding site and promoter-proximal region of class I TSG (*Figure 3A–C*), in perfect agreement with the Tat-mediated increase in H3K4me3 at those gene promoters (*Figure 2A,B*). It is noteworthy that this observation is consistent with the model that Tat mediates transcription initiation at class I TSG. Conversely, as expected, we did not see higher levels of Pol II at the promoters of class II TSG in response to Tat (*Figure 3A,B,D*). This lack of Pol II recruitment further supports our proposed model based on the global analysis of chromatin signatures that class II TSG are not regulated at the initiation step (*Figure 2*).

Interestingly, a metagene analysis of class I TSG in the absence of Tat showed very low Pol II density levels in both the promoter and transcribing unit. Conversely, Tat induces an increase in Pol II density throughout the gene body with a noticeable peak at the promoter-proximal region, consistent with Pol II recruitment to promoters and transition into elongation (*Figure 3E*) and with the increase in H3K4me3 at the promoter-proximal regions associated with those genes (*Figure 2A*, and *Figure 2—figure supplement 3*). Genome browser views of individual class I TSG such as *NETO1* exemplify the metagene analysis depicting that Pol II is strongly recruited to the intragenic Tat binding site and promoter, marked with H3K4me3 in the presence of Tat (*Figure 3F*), and that transcription initiation correlates with increase in chromatin signatures (H3K79me3 and H3K36me3) associated with active transcription in gene bodies, as well as transcribing Pol II (*Figure 3—figure supplement 1A*).

We have previously defined class II TSG as being primarily regulated at the elongation step because H3K4me3 and Pol II density at the promoter-proximal region do not increase in the presence of Tat (*Figure 2* and *3A–D*). Examination of two class II TSG (*VAV3* and *CD82*) using Chip-qPCR clearly demonstrates that Tat induces Pol II elongation by recruiting the P-TEFb kinase at those target genes (*Figure 3—figure supplement 2A,B*). Consistently, a metagene analysis shows that Tat largely increases Pol II density throughout the gene body and 3′-end of class II TSG, but not at the promoter, in agreement with a role of Tat in promoting the transition to elongation (*Figure 3G*). Therefore, in the absence of Tat stimulation, the majority of Pol II at class II TSG accumulates in the promoter-proximal region with a peak just downstream of the TSS while Tat strongly induces increased Pol II density in the gene body but not in the promoter-proximal region (*Figure 3G*). Genome browser views of individual class II TSG such as *VAV3* are consistent with the elongation function (*Figure 3H*), and are also in perfect agreement with the analysis of chromatin signatures related to elongation (H3K79me3 and H3K36me3) (*Figure 2* and *Figure 3—figure supplement 1B*).

Collectively, the data indicate that Tat controls both Pol II recruitment and pause release to promote initiation and elongation in different gene classes.

## Tat stimulates transcription initiation from intragenic enhancers by inducing gene looping

Tat is recruited to two different genomic domains (promoters and intragenic) irrespective of the transcription step regulated (initiation or elongation) (*Figure 4A*), implying that the site of Tat recruitment to its target genes does not dictate the mechanism of transcription activation. To further elucidate how Tat promotes transcription by binding to promoters or intragenic sites, we examined how chromatin signatures associated with these genomic domains change in response to Tat. Promoters bound by Tat are marked with the expected signature: high H3K4me3 and low H3K27Ac (*Figure 4B*) (*Creyghton et al., 2010*; *Heintzman and Ren, 2009*). Conversely, intragenic sites bound by Tat are marked with the enhancer signature: high H3K27Ac and low H3K4me3 (*Figure 4B*), as well as high H3K4me1 (data not shown) (*Creyghton et al., 2010*; *Heintzman and Ren, 2009*; *Zhou et al., 2011*). This suggests that the intragenic Tat-binding sites appear to be intragenic enhancers, which have been previously proposed to function as alternative elements required for gene activation (*Kowalczyk et al., 2012*).

To better define the roles of Tat in activating transcription from these promoter-distal, intragenic sites, we sorted class I and II TSG based on the H3K27Ac density surrounding the Tat peak. Notably, we observed that in the absence of Tat, the intragenic sites at class I TSG have low or undetectable levels of H3K27Ac (*Figure 4C*), consistent with the idea that these genes are inactive or only minimally transcribed in the basal state without Tat (*Figure 2*). However, Tat increases H3K27Ac density

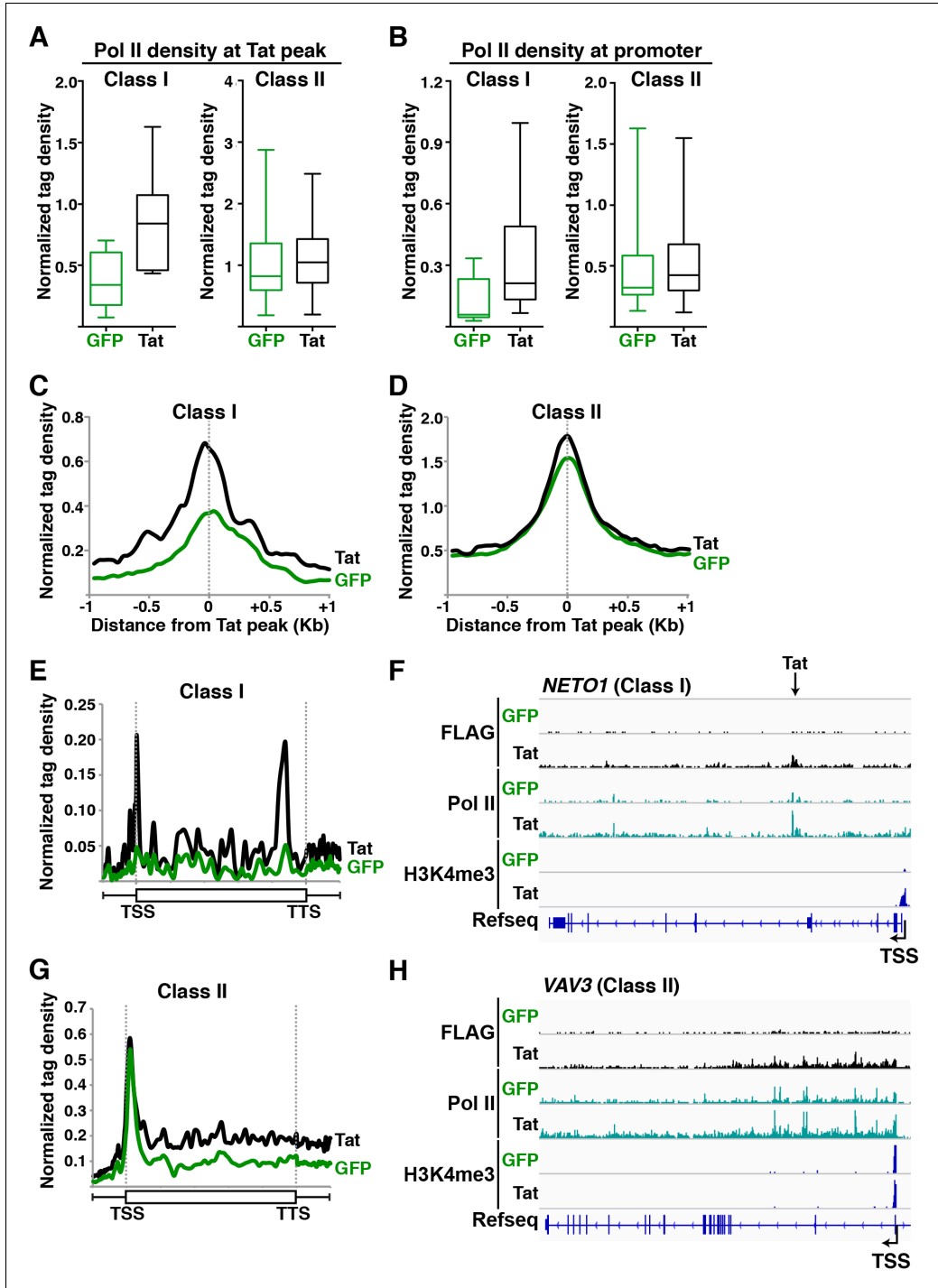

**Figure 3.** Tat promotes Pol II recruitment and pause release at two distinct TSG classes. (A) Tat binding promotes Pol II recruitment at class I but not class II TSG. Box plots of normalized Pol II tag density at the Tat peak of class I (n = 17) or II (n = 43) TSG in the GFP and Tat cell lines. (B) Tat binding at class I TSG induces Pol II occupancy at promoters. Box plots of normalized Pol II tag density at the promoter of class I or II TSG in the GFP and Tat cell lines. (C) Pol II normalized tag density relative to the Tat peak at class I TSG. (D) Pol II normalized tag density relative to the Tat peak at class II TSG. (E) Pol II distribution at class I TSG (Metagene plots) in the Tat and GFP cell lines. (F) Genome browser views of ChIP-seq data in the Tat and GFP cell lines at a class I TSG (*NETO1*). The arrow indicates the position of the FLAG peak called in the Tat cell lines by MACS. (G) Pol II distribution at class II TSG (Metagene plots) in the Tat and GFP cell lines. (H) Genome browser views of ChIP-seq data in the Tat and GFP cell lines at a class II TSG (*VAV3*). This figure is associated with *Figure 3—figure supplements 1,2*. ChIP-seq, chromatin immunoprecipitation sequencing; GFP, green fluorescent protein; MACS, model-based analysis of ChIP-seq; Pol, polymerase; TSG, Tat stimulated genes.

*Figure 3 continued on next page*

*Figure 3 continued*

The following figure supplements are available for figure 3:

**Figure supplement 1.** Tat recruitment induces Pol II and chromatin signatures controlling transcription initiation or elongation at different gene classes.

**Figure supplement 2.** Tat controls P-TEFb and Pol II recruitment at class II TSG and class II TDG.

near its intragenic binding sites by ~2–10-fold depending on the gene (*Figure 4C,D*). Conversely, in the absence of Tat, H3K27Ac levels at the intragenic sites of class II TSG are high and Tat increases their density, albeit with a lower fold-change than in class I TSG (*Figure 4C,D*). This is consistent with the model that these genes are active in the basal state and Tat activates a post-initiation step, namely transcription elongation. Metagene analysis of class I TSG indicates that the H3K27Ac mark at intragenic binding sites is virtually absent at the immediate binding site itself but high in the nucleosomes directly surrounding the Tat peak, progressively declining further up and downstream from the binding site (*Figure 4E*). At these sites, Tat increased H3K27Ac levels an average of ~3-fold. A similar H3K27Ac distribution pattern is observed in class II TSG, even though the magnitude of H3K27Ac increase is smaller (~1.5-fold) because these genes are already active in the absence of Tat (*Figure 4F*).

Given that the histone acetyl transferase p300/CBP is a well-known Tat interactor and that it is recruited to enhancers to facilitate transcription activation through chromatin acetylation (including H3K27Ac) (*Hottiger and Nabel, 1998*; *Jager et al., 2012*; *Kim et al., 2010b*), we asked whether Tat recruits p300 to these intragenic sites to trigger H3K27Ac. To test this possibility, we performed ChIP assays on one class I TSG (*PPM1H*) and class II TSG (*CD82*). Notably, we observed that in the presence of Tat, the increase in H3K27Ac levels at class I intragenic sites in response to Tat mirrors the recruitment of p300 (*Figure 4G*). However, levels of both H3K27Ac and p300 detected at the intragenic sites of class II TSG are already high in the basal state and are slightly induced (~1.2–1.4-fold) in response to Tat (*Figure 4H*).

Remarkably, we observed a sharp correlation ($R^2$ = 0.853) between increased H3K27Ac density at the intragenic site and gene expression levels at class I TSG (*Figure 4—figure supplement 1A*), supporting a model in which Tat is recruited to these sites to induce de novo transcription activation through recruitment of p300 at internal sites. However, the correlation between H3K27Ac density at the intragenic site and gene expression levels at class II TSG is quite low ($R^2$ = 0.0728) because these genes are already marked by H3K27Ac in the absence of Tat (*Figure 4—figure supplement 1B*). These results support the model that Tat binds at intragenic sites of non-productive genes (class I TSG) to induce their transcription initiation or is recruited to productive genes (actively transcribed) such as class II TSG to augment transcription elongation levels.

Because these Tat binding sites are located distally from the promoter in the majority of class I TSG (15 out of 17), we hypothesized a model in which Tat controls gene looping to induce spatial proximity between the intragenic site (putative enhancer) and the promoter. To further test this possibility in detail we selected one class I TSG (*PPM1H*), which is about 300 kb in length, thus facilitating the analysis of long-range chromatin interactions (*Figure 4I*). We performed chromosome conformation capture (3C) to assess the association of two intragenic Tat target sites and the promoter. We also performed a similar analysis between the gene promoter and distal intergenic sites located at a similar distance from the promoter but not bound by Tat. We used anchoring points near the gene promoter to measure the extent of chromatin looping between the promoter and the intragenic Tat-bound sites or the distal control domain. Interestingly, this analysis revealed that prior to Tat stimulation there is no obvious interaction between the gene promoter and the two intragenic Tat-bound sites. However, Tat promotes a strong association between the two sites, albeit with different crosslinking efficiencies (*Figure 4I*), and this gene looping needs a functional Tat, because the C22A non-functional Tat mutant does not promote this effect (*Figure 4—figure supplement 2*). Moreover, the effect of Tat on chromatin looping is specific as there was no detectable association when a control anchor primer was placed about 20 kb upstream from the *PPM1H* promoter (*Figure 4—figure supplement 2*).

Gene looping could be a direct consequence of Tat activity at promoters and enhancers, or Tat can simply help activate enhancers, causing them to increase looping/proximity to their target

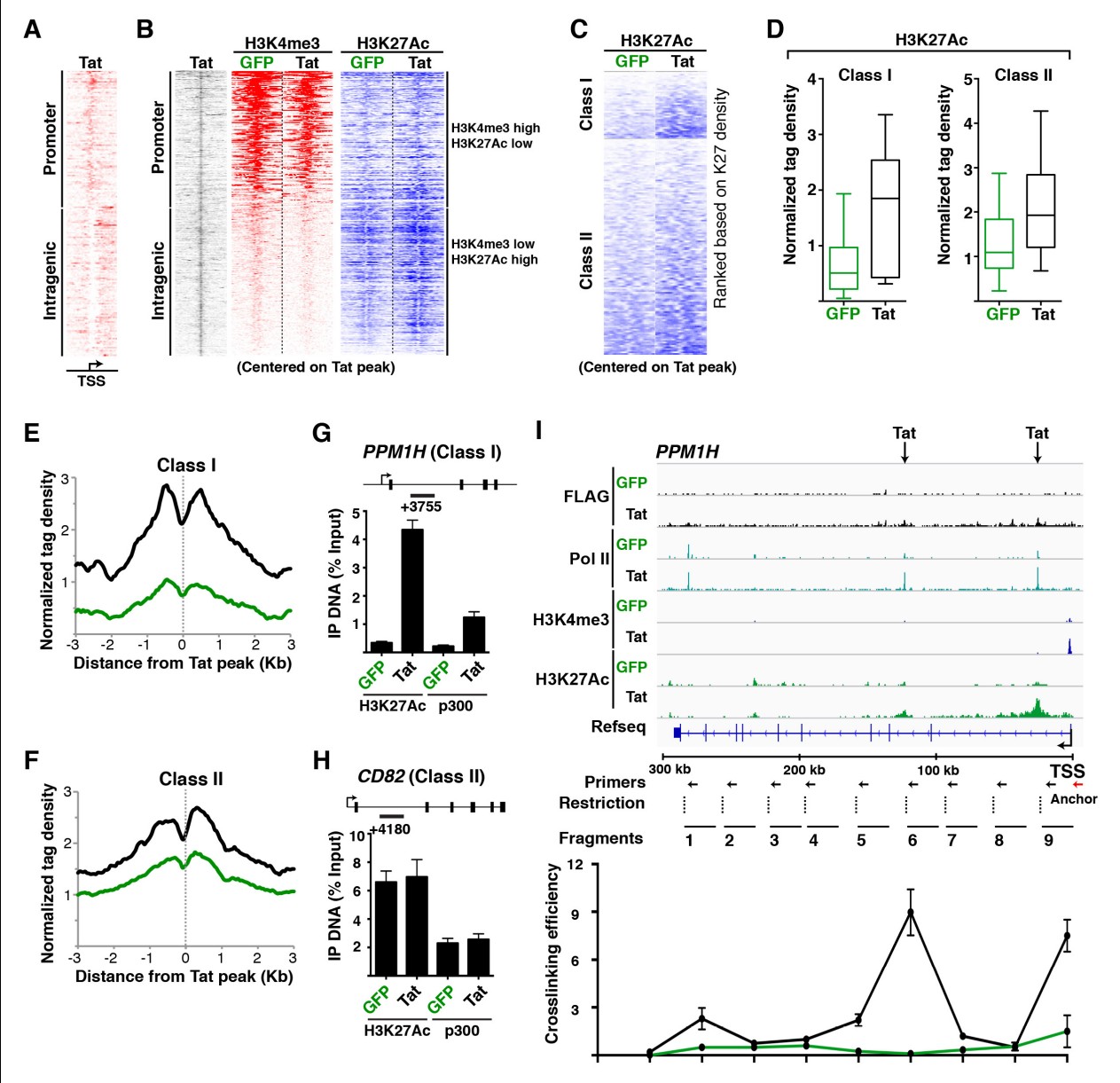

**Figure 4.** Tat induces transcription initiation from distal sites by inducing gene looping. (**A**) Heatmap representation of Tat distribution at TSG promoter or intragenic sites. (**B**) Heatmap representation of H3K4me3 and H3K27Ac tag density centered on the Tat peak at both TSG promoter and intragenic sites in the GFP and Tat cell lines. Promoter sites are marked with high H3K4me3 and low H3K27Ac levels, while intragenic sites are marked by low H3K4me3 and high H3K27Ac levels. (**C**) Heatmap representation of H3K27Ac at class I and class II TSG centered on the Tat peak in the GFP and Tat cell lines. Genes are ranked based on H3K27Ac density. (**D**) Tat binding sharply increases H3K27Ac levels at class I TSG. Box plots showing normalized H3K27Ac density at class I and II TSG in the GFP and Tat cell lines. (**E**) Metagene analysis of H3K27Ac levels surrounding Tat peaks at class I TSG in the GFP and Tat cell lines. (**F**) Metagene analysis of H3K27Ac levels surrounding Tat peaks at class II TSG in the GFP and Tat cell lines. (**G**) ChIP of H3K27Ac and p300 recruitment at an intragenic Tat site at the *PPM1H* gene. (**H**) ChIP of H3K27Ac and p300 recruitment at an intragenic Tat site at the *CD82* gene. (**I**) Top, genome browser views of ChIP-seq at the *PPM1H* locus in the GFP and Tat cell lines. The arrows indicate the position of two intragenic Tat binding sites. Bottom, 3C assay showing the relative crosslinking efficiency at the *PPM1H* locus in the GFP and Tat cell lines. The position of the primers used in qPCR assays, restriction sites and fragment generated after digestion are indicated. This figure is associated with *Figure 4—figure supplements 1–4*. 3C, chromosome conformation capture; ChIP-seq, chromatin immunoprecipitation sequencing; GFP, green fluorescent protein; RNA-seq, RNA sequencing; TSG, Tat stimulated genes.

The following figure supplements are available for figure 4:

**Figure supplement 1.** Stimulation of gene expression by Tat correlates with increased H3K27Ac density at the Tat peak.

*Figure 4 continued on next page*

*Figure 4 continued*

**Figure supplement 2.** Wild-type Tat but not the C22A non-functional mutant induces gene looping between the promoter and intragenic sites.

**Figure supplement 3.** Tat-mediated gene looping does not require transcription activity at enhancers.

**Figure supplement 4.** Class II TSG contain stable gene loops between promoter-enhancer that remain unaltered in response of Tat.

promoters (*Bulger and Groudine, 2011*). To distinguish between these two possibilities we used flavopiridol (FP), a CDK9 inhibitor that, in addition to blocking transcription elongation, also inhibits the production of enhancer-derived RNAs (eRNAs), without affecting other molecular indicators of enhancer activity (such as Pol II binding, H3K4me1 levels, and gene looping) (*Hah et al., 2013*). We used FP in the Jurkat-GFP and -Tat cell lines and performed 3C experiments to uncouple the assembly of enhancer complexes and gene looping per se from eRNA production. The data suggests that Tat promotes gene looping even in the absence of eRNA synthesis (*Figure 4—figure supplement 3A–D*), implying that eRNA synthesis occurs after the assembly of Tat and the transcription machinery at the enhancer, and enhancer-promoter communication.

We have suggested that class II TSG have paused Pol II, and it is known that promoter-enhancer loops are associated with paused Pol II (*Ghavi-Helm et al., 2014*). We thus wished to test whether gene looping also takes place at class II TSG, and if Tat plays any role. We first analyzed the percentage of class II TSG-containing intragenic Tat-bound sites (n = 34 out of 43, 79%) and then examined whether gene looping is critical for their expression and whether Tat promotes the looping (*Figure 4—figure supplement 4*). For this we selected one class II TSG (*CD82*), which shows a strong Pol II peak promoter-proximally, has one Tat-bound site intragenically, and shows evidence of looping between the promoter and the intragenic Tat-bound site. We found no evidence that Tat modulates gene looping at the *CD82* loci thereby indicating that Tat's effects at class II TSG is at a step post-formation of long-range chromatin interactions (*Figure 4—figure supplement 4*), in agreement with a role of Tat in promoting elongation.

In conclusion, class I TSG is transcriptionally inactive (or transcribed at a low rate) and Tat binds to intragenic sites to induce chromatin looping and transcription activation. On the other hand, class II TSG is actively transcribed and Tat promotes a post-initiation step (namely Pol II elongation) by binding to both promoters and/or intragenic sites without affecting gene looping.

## Implementation of a modified traveling ratio algorithm indicates that Tat promotes transcription elongation at class II TSG

While at class I TSG Tat promotes the initiation step, at class II TSG, Tat appears to induce Pol II transition into the elongation phase (*Figures 2,3*). Previously, an algorithm that computes Pol II levels in promoter-proximal regions and gene body (termed Traveling Ratio [TR]) has been devised to examine the transition from initiation into elongation (*Figure 5A*) (*Rahl et al., 2010*; *Reppas et al., 2006*; *Wade and Struhl, 2008*; *Zeitlinger et al., 2007*). We thought to apply such an algorithm to examine Tat functions in initiation and elongation by comparing Pol II levels found in promoter-proximal regions (-50 to +1000 respective to the TSS) versus levels within the gene body. Unexpectedly, we found in our ChIP-seq datasets that in both the presence and absence of Tat, Pol II levels tend to peak not only near the promoter of many genes, but also at certain intronic domains marked by H3K27Ac and H3K4me1, sometimes at densities similar to those found at the promoter-proximal region (*Figure 5B*). These sites do not contain any known annotation in the University of California Santa Cruz (UCSC) genome browser and does not appear to contain a non-canonical TSS because the H3K4me3 levels are low-to-undetectable. Thus, to examine if these Pol II forms were technical artifacts, we performed ChIP-seq with other Pol II Abs (total Pol II, CTD, Ser5P-CTD and Ser2P-CTD), both in the absence and presence of Tat, and found that, irrespective of the Ab used, this stably paused Pol II form was detected in the gene body, albeit with variable density levels (*Figure 5—figure supplement 1*). Similarly, genome browser searches revealed that intragenically poised Pol II is also detected in primary CD4+ T cells, thus ruling out the possibility that it is an artifact of Pol II distribution present in immortalized T cell lines growing in tissue culture. Moreover, examination of

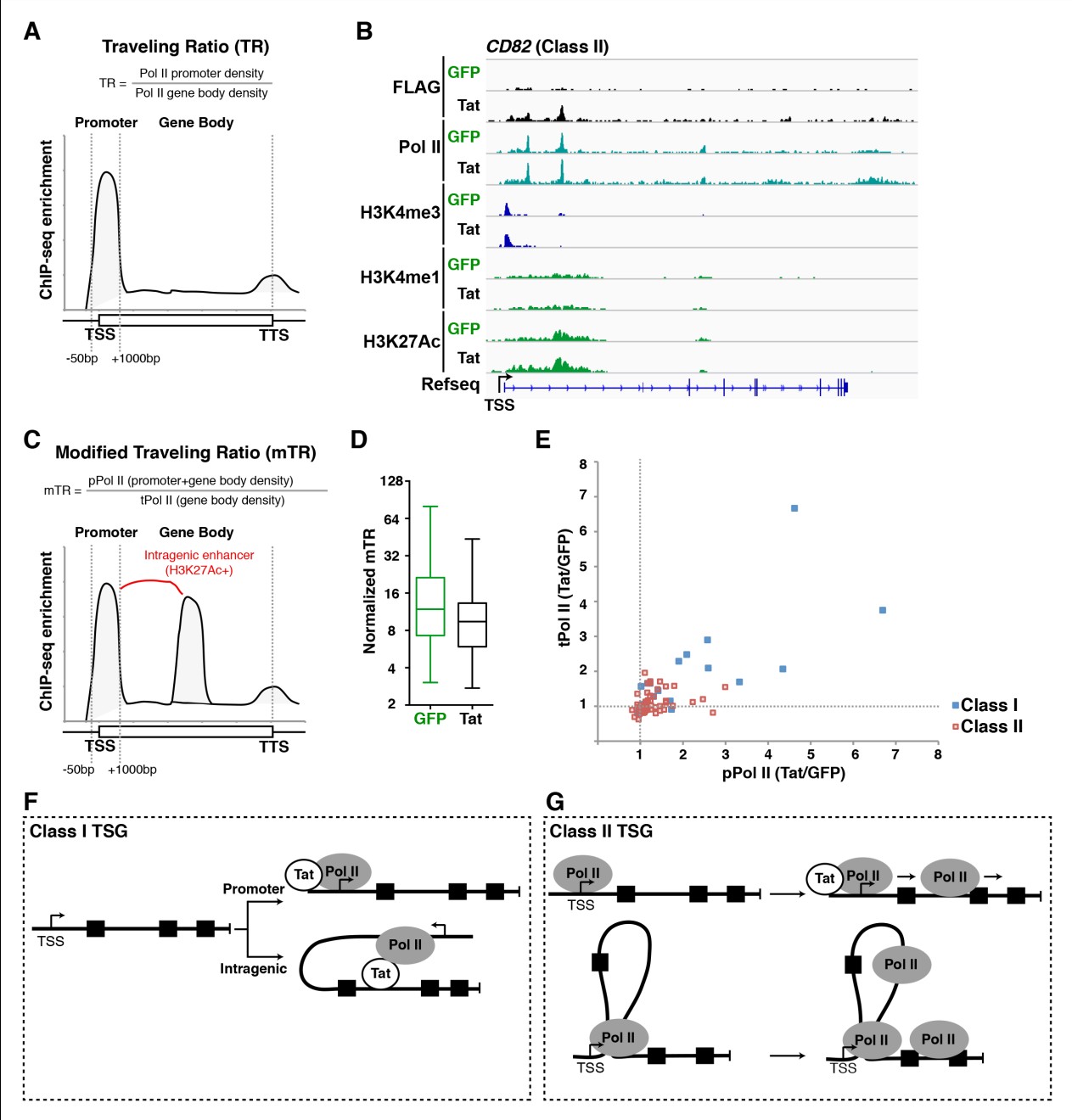

**Figure 5.** A modified traveling ratio algorithm reveals Tat roles in transcription elongation. (**A**) TR algorithm used to calculate levels of promoter-proximal paused and elongation Pol II as previously described (*Rahl et al., 2010*; *Zeitlinger et al., 2007*). (**B**) Genome browser views of ChIP-seq data at the *CD82* locus in the GFP and Tat cell lines. (**C**) mTR algorithm that enables the identification of intragenically paused Pol II at sites of high H3K27Ac levels. (**D**) mTR decreases at TSG in the Tat cell line. Box plots showing mTR calculations at TSG as indicated in panel (**C**). (**E**) Dual-axis plot of the ratio of paused Pol II (pPol II) and traveling Pol II (tPol II) at class I (n = 17) and II (n = 43) TSG in the presence and absence of Tat (Tat/GFP). (**F**) Model of Tat-mediated Pol II recruitment and gene looping at class I TSG. Tat binds at promoter and/or intragenic regions (in some cases inducing gene looping), and recruits Pol II to promote transcription initiation from both promoter-proximal and promoter-distal sites. (**G**) Model of Tat-mediated Pol II elongation at class II TSG. Class II TSG are already activated in the absence of Tat, and Tat binds to these genes to primarily stimulate transcription elongation. These genes contain promoter-bound Pol II. This figure is associated with *Figure 5—figure supplement 1*. GFP, green fluorescent protein; Pol, polymerase; mTR, modified TR; TR, traveling ratio; TSG, Tat stimulated genes.

The following figure supplement is available for figure 5:

**Figure supplement 1.** Intragenically paused Pol II is detected in CD4+ T cell lines and primary cells at sites with high H3K27Ac and low H3K4me3.

our ChIP-seq chromatin signature tracks indicated that the sites of intragenic Pol II pausing correspond to potential enhancers because they are marked with the corresponding enhancer signature (high H3K4me1/H3K27Ac and low H3K4me3) (*Figure 5B*) (*Heintzman and Ren, 2009*).

Given that we found intragenic enhancers containing a form of Pol II that appears to be paused, and that enhancer–promoter interactions are frequently associated with paused Pol II (*Ghavi-Helm et al., 2014*), we speculated that the original TR algorithm might not accurately describe Tat's role in transcription elongation in our dataset. The problem arises because the TR algorithm treats all Pol II in the gene body as actively elongating. This assumption is invalid in cases where Pol II appears paused intragenically or is detected at an intragenic enhancer because of chromatin looping. In order to circumvent this problem we devised a custom-made algorithm referred to as 'modified TR' (mTR) to identify intragenic enhancers marked by H3K4me1/H3K27Ac and paused Pol II. mTR accurately categorizes these sites to more precisely calculate the promoter-proximal and gene body Pol II counts (*Figure 5C*). We defined paused Pol II (pPol II) as Pol II located in both the promoter-proximal region (-50/+1000 from the TSS) and at intragenic sites marked with H3K27Ac (as determined by MACS2, with a peak cutoff of p-value <0.05) and having a read density/nt greater than five times that of the average density/nt within the gene body. Of note, we used a window of -50/+1000 from the TSS (rather than -50/+300) because several genes contained accumulation of Pol II beyond the +100, and up to the +1000 position. Transcribing Pol II (tPol II) is defined as Pol II in the remainder of the gene that is not associated with the enhancer signature. Then, the mTR is the ratio of average pPol II (promoter + gene body) density to average tPol II (gene body) density (*Figure 5C*). As expected, class II TSG showed decreased mTR in the presence of Tat compared with the GFP cell line (*Figure 5D*).

Despite this compelling observation, the mTR could be altered in response to Tat either by fluctuations in the levels of pPol II, tPol II, or both. Therefore, to more clearly explore Tat's role in controlling elongation, we computed densities for both Pol II classes (pPol II and tPol II) and compared them separately in the GFP and Tat cell lines. For each gene, we calculated the ratio of average pPol II and tPol II densities in the two cell lines (Tat/GFP) and plotted them on the X- and Y-axis, respectively (*Figure 5E*). This plot demonstrates clearly the Pol II behavior in response to Tat, both in terms of changes to initiation and elongation. While class I TSG showed profound increase in pPol II and tPol II as a consequence of transcription initiation/elongation activation, class II TSG showed, albeit with gene-specific differences, higher evidence of elongation (increased tPol II form in the presence of Tat) (*Figure 5E*).

Together, the data suggest that Tat has evolved to fine-tune both transcription initiation and elongation steps at functionally different gene classes. At class I TSG, Tat directly mediates Pol II recruitment to gene promoters or binds to intragenic sites to induce chromatin looping (promoter-enhancer communication) to trigger transcription initiation (*Figure 5F*). At class II TSG, Tat binds to already engaged Pol II to promote its escape into the productive elongation phase (*Figure 5G*).

## Global analysis of chromatin signatures and Pol II distribution indicate that Tat downregulates transcription by blocking both initiation and elongation

Global analysis of chromatin signatures in stimulated genes revealed that Tat induces both transcription initiation and elongation (*Figure 2*). Thus, we computationally examined chromatin signatures throughout promoter-proximal and -distal regions of TDG to determine at which step in the transcription cycle Tat blocks gene activation. Despite the identification of genes having marked changes in both chromatin initiation (H3K4me3) and elongation signatures (H3K79me3 and H3K36me3) such as *CD1E* and *FBLN2*, or only chromatin elongation signatures such as *EOMES* and *HSPA8*, this distinction did not become as clear for genes having low gene expression changes (<3-fold). Therefore, we used a combined chromatin/Pol II signature to identify initiation- (class I) and elongation-regulated (class II) TDG (*Figure 6*). We defined class I TDG as genes having chromatin and Pol II signatures consistent with inhibited transcription initiation, with at least 60% decrease in H3K4me3 surrounding the TSS and 30% decrease in promoter-proximal Pol II with Tat (fold H3K4me3 $_{Tat/GFP}$<0.4 and fold Pol II$_{Tat/GFP}$<0.7). Conversely, class II TDG are inhibited at later stages of transcription because H3K4me3 and Pol II promoter-proximal levels remain virtually unchanged in the presence of Tat (fold H3K4me3 $_{Tat/GFP}$>0.7 and fold Pol II$_{Tat/GFP}$>0.75). Therefore, we defined class II TDG as genes having less than 30% decrease H3K4me3 surrounding the TSS and less than

25% decrease in promoter-proximal Pol II level in response to Tat. Using this signature, we identified 11 class I TDG and 14 class II TDG that strictly pass the above-mentioned criteria. Remarkably, these genes also appear to be controlled at the initiation and elongation level in primary T$_{CM}$ cells infected with HIV (*Figure 1—figure supplement 8E,F*). Although further research is needed to pinpoint the details, the rest of the genes could be a mixed class I-II population in which Tat simultaneously interfere with both the initiation and elongation steps (see Discussion).

Class I TDG experience a sharp decrease in H3K4me3 (~4.2-fold change) and in Pol II (see below) in the presence of Tat (*Figure 6A,B*), consistent with a role of Tat in blocking transcription initiation. In support of this view, we observed a statistically significant correlation between the decrease in H3K4me3 levels surrounding the TSS and RNA levels at class I TDG but not class II (data not shown). To further examine how Tat blocks the initiation step, we performed detailed ChIP-qPCR assays to monitor the occupancy of subunits of the transcription pre-initiation complex (PIC). Interestingly, we observed that Tat recruitment to the *CD1E* gene promoter blocks the step of PIC assembly as denoted by the sharp decrease in the occupancy levels of TBP, Mediator (MED1), Pol II and the P-TEFb kinase (CDK9) (*Figure 6—figure supplement 1A*), in agreement with the virtual removal of H3K4me3 (*Figure 6A,G*). On the contrary, class II TDG showed no significant change in H3K4me3 density surrounding the TSS (*Figure 6B,H*), which is in agreement with the lack of Tat effects at the transcription initiation level. Consistently, at the class II TDG *EOMES*, Tat does not interfere with PIC assembly (as revealed by identical levels of TBP and MED1 at the promoter). However, Tat blocks Pol II transition into productive elongation due to dismissal of P-TEFb from both the promoter and transcription unit (*Figure 6—figure supplement 1B*), which affects levels of total Pol II and the elongating form (Ser2P) in the gene body but not at the promoter-proximal region at two class II TDG (*EOMES* and *HSPA8*) (*Figure 3—figure supplement 2C,D*).

If Tat blocks transcription initiation, then we would predict that the chromatin elongation signatures would also decrease, as there is no recruited Pol II available for elongation. In agreement, class I TDG showed a virtual elimination of H3K79me3 and H3K36me3 levels in the transcribed unit, reflecting potent inhibition of these genes (*Figure 6C–F*). The metagene analysis clearly indicated that all class I TDG showed reduced levels of both chromatin elongation signatures (*Figure 6D,F*). However, the difference in magnitude at different genes was very broad, most likely due to the fact that these genes are transcribed at different levels and their abundance (based on RNA-seq) is quite disparate (*Figure 6C,E*). Genome browser inspection of individual genes such as *CD1E*, which is expressed at high levels, provide evidence that Tat is a very potent transcription initiation repressor, as revealed by the large drop in all chromatin signatures profiled (*Figure 6G*) and inhibition of PIC assembly at the promoter, as revealed by the dismissal of TBP, MED1, CDK9 and Pol II at the *CD1E* loci (*Figure 6—figure supplement 1A*).

In contrast to class I TDG, class II showed no significant change in H3K4me3 levels surrounding the TSS (*Figure 6A,B*). However, a subset of class II genes such as *EOMES*, *HSPA8* and *CDK6*, showed reduced chromatin elongation signatures (H3K79me3 and H3K36me3) throughout the transcribing unit (*Figure 6C,E,H*). Genome browser inspection of individual class II TDG such as *EOMES* demonstrate that Tat blocks the transition to elongation because Pol II density largely decreases at the transcribing unit without significant alterations in the promoter-proximal region, consistent with decreased chromatin elongation signatures but no H3K4me3 effects (*Figure 6H*). However, surprisingly, a few genes showed no change in H3K79me3 modification, or even a small increase, right after the TSS (*Figure 6C*), possibly related to the fact that Dot1L-mediated establishment of H3K79me3 also has been linked with transcriptional repression (*Nguyen and Zhang, 2011*; *Onder et al., 2012*). However, this alternative function of Dot1L in transcription repression will require further investigation.

Taken together, it is evident that Tat functions as a transcriptional repressor blocking Pol II recruitment and pause release as well as promoting the removal of chromatin modifications associated with initiation and elongation at different gene classes.

## Tat blocks Pol II recruitment and pause release to repress transcription initiation and elongation

The fact that Tat modifies the epigenetic landscape to repress cellular transcription prompted us to examine Tat effects on Pol II distribution changes (*Figure 7*). If Tat represses transcription initiation by blocking PIC assembly such as in the *CD1E* gene then we would expect Pol II levels to decrease

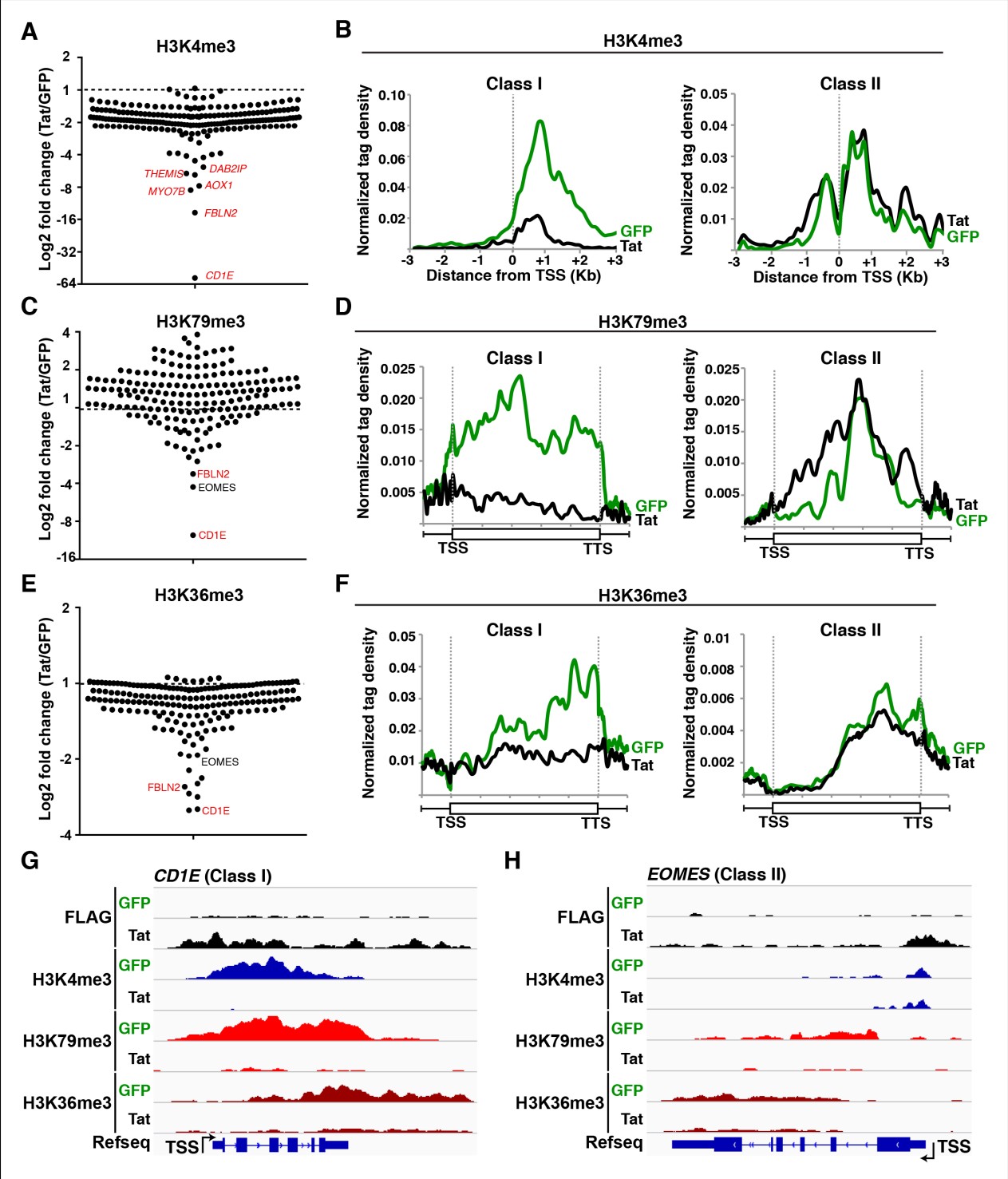

**Figure 6.** Global analysis of chromatin signatures reveals the basis of transcriptional repression by Tat. (**A**) Dot plots of H3K4me3 log2 fold change in the region encompassing -3/+3 Kb from the TSS of all TDG. TDG are divided into two classes: class I (n = 11) and II (n = 14) based on decreased or unchanged levels of H3K4me3 in the presence of Tat, respectively. Selected TDG are indicated in red. (**B**) Metagene plots centered on TSS showing H3K4me3 occupancy profiles at both class I and class II TDG in the presence of Tat or GFP. (**C**) Dot plots of H3K79me3 log2 fold change from TSS to TTS. (**D**) Metagene plots showing average H3K79me3 ChIP-seq signals at both class I and class II TDG in the presence of Tat or GFP (see Materials and methods). (**E**) Dot plots of H3K36me3 log2 fold change in the region from TSS to TTS. (**F**) Metagene plots showing average H3K36me3 ChIP-seq signals at both class I and class II TDG in the presence of Tat or GFP. (**G**) Genome browser views showing ChIP-seq signal at a class I TDG (*CD1E*) in the GFP and Tat cell lines. (**H**) Genome browser views showing ChIP-seq signal at a class II TDG (*EOMES*) in the GFP and Tat cell lines. This

*Figure 6 continued*

figure is associated with *Figure 6—figure supplement 1*. ChIP-seq, chromatin immunoprecipitation sequencing; GFP, green fluorescent protein; TDG, Tat downregulated genes; TSS, transcription start site; TTS, transcription termination site.

The following figure supplement is available for figure 6:

**Figure supplement 1.** Different modes of Tat repression at class I and II TDG.

at gene promoters and other functionally relevant sites in the presence of Tat. We first analyzed levels of Pol II at both Tat binding sites (Tat peak) and promoters of associated genes for both class I and II TDG. Interestingly, we found that Tat blocks Pol II recruitment at both the Tat binding site and promoter-proximal region of class I TDG (*Figure 7A–C*), in perfect agreement with the Tat-mediated decrease in H3K4me3 at those promoters (*Figure 6*). Remarkably, these observations are in agreement with the model that Tat blocks transcription initiation at class I TDG. Conversely, as expected based on the global analysis of chromatin signatures, Tat does not appear to largely block Pol II recruitment at the Tat peak and promoter of class II TDG (*Figure 7A,B,D*), which we proposed to be regulated at the elongation level based on the global analysis of chromatin signatures and Pol II levels (*Figure 6*). In fact, this is in agreement with the lack of H3K4me3 density changes at promoters of class II TDG in the presence of Tat.

Metagene analysis of class I TDG showed high Pol II density levels in both the gene and transcribing unit in the absence of Tat. However, Tat reduces Pol II density at the promoter, consistent with a drop in transcribing Pol II at the gene body (*Figure 7E*). Genome browser views of individual class I TDG such as *CD1E* clearly exemplify the metagene analysis depicting that Pol II recruitment at the promoter is strongly blocked, leading to the virtual disappearance of Pol II signal throughout the gene body. The transcription initiation signature H3K4me3 is also eliminated (*Figure 7F*), and the blockage of transcription initiation correlates with decrease in chromatin signatures associated with transcription elongation (H3K79me3 and H3K36me3) throughout the gene body (*Figure 6*).

We have defined class II TDG as being primarily controlled at the elongation step because H3K4me3 and Pol II density at the promoter-proximal region does not appear to be modified by Tat (*Figures 6* and *7*). Consistently, a metagene analysis shows that Tat decreases Pol II density throughout the gene body and 3'-end, but not at the promoter-proximal region, in agreement with a role of Tat in blocking the transition to elongation (*Figure 7G*). Genome browser views of individual class II TDG such as *EOMES* are consistent with the block of Pol II pause release (*Figure 7H*), and with the global analysis of chromatin signatures related to elongation (*Figure 6*). Reduced elongation is due to the block of P-TEFb recruitment to promoters in class II TDG (*EOMES* and *HSPA8*) (*Figure 3—figure supplement 2*). Moreover, simultaneous analysis of the poised and transcribing Pol II forms (pPol II and tPol II) using our mTR algorithm provides further evidence that, in the majority of class II TDG, Tat dampens the level of tPol II without largely modifying or even increasing pPol II (*Figure 7—figure supplement 1*), consistent with a role in primarily blocking the transition into elongation.

In conclusion, the data indicate that Tat precisely modulates Pol II recruitment and transition into the gene body to block transcription initiation or elongation, respectively, at different gene classes.

## Enrichment of motifs for master transcriptional regulators but not of TAR-like motifs in the Tat binding sites

Given that Tat activates transcription from the HIV promoter by associating with the TAR structure that is formed at the nascent chain of viral pre-mRNAs (*Frankel and Young, 1998*) (*Figure 8A*), we investigated the possibility that Tat is recruited to the host chromatin through interaction with TAR-like structures. To test whether the Tat sites at the direct target genes (both TSG and TDG) contain TAR-like structures, we searched for TAR-like motifs using a custom algorithm and the input sequence $X_{(2)}GATX_{(1,2)}GAX_{(4,40)}TCTCX_{(2)}$ as query (*Figure 8B*), where X denotes any nucleotide, and the numbers in brackets represent the minimal and maximal allowed positions with the corresponding secondary structure pattern $XX((XX((X_{(4-40)}-X))))XX$ including the di-/tri-nucleotide bulge within the stem and loop (both critical determinants for TAR binding) (*Frankel and Young, 1998*). Locations of TAR-like motifs near target genes were cataloged and compared to distribution of Tat peaks in our ChIP-seq dataset. While ~20% of the TSG and TDG contain TAR-like motifs within a very large

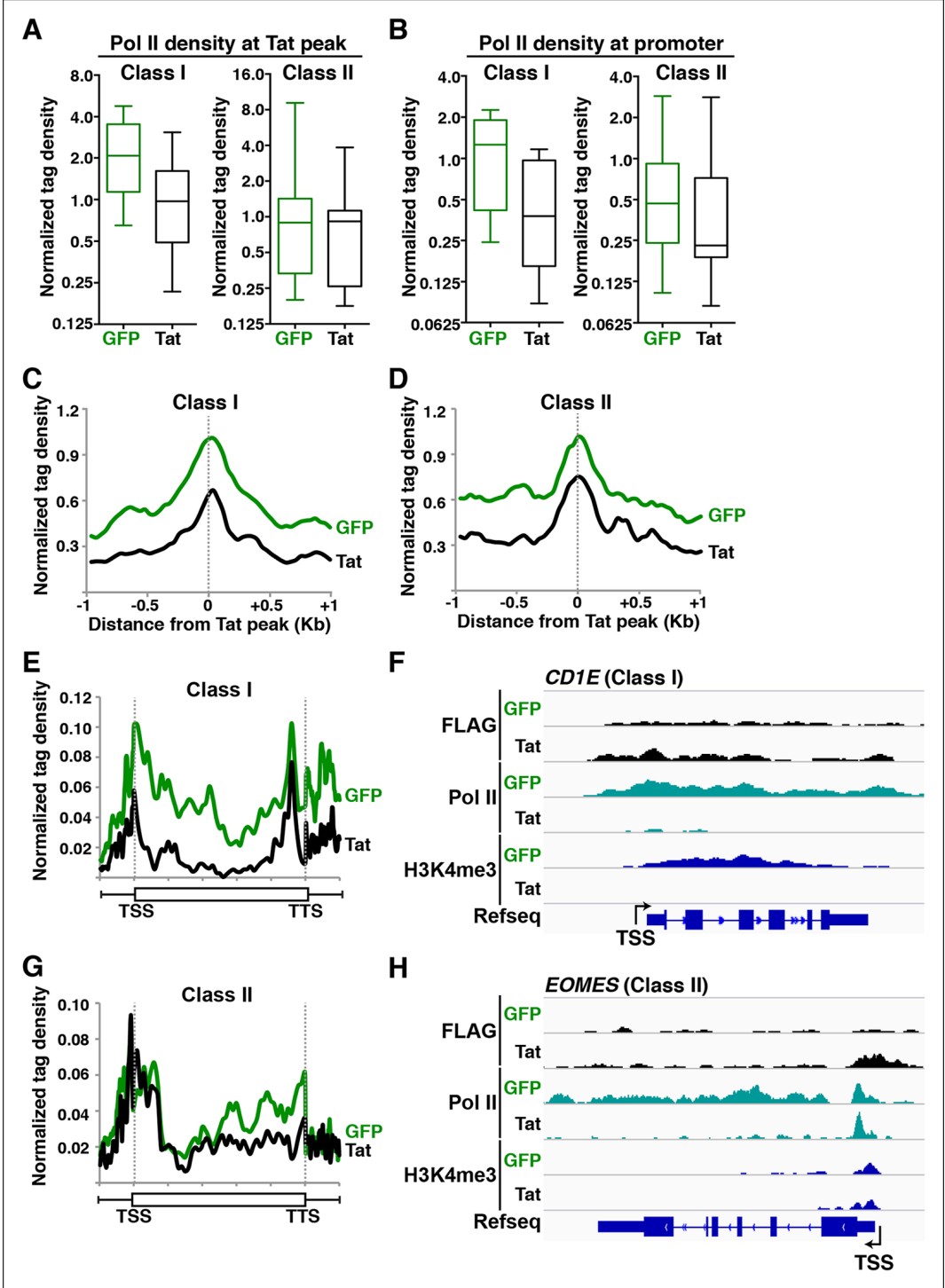

**Figure 7.** Tat blocks Pol II recruitment and pause release to repress gene transcription at different gene classes. (**A**) Tat binding blocks Pol II recruitment primarily at class I TDG. Box plots of normalized Pol II tag density at the Tat peak of class I or II TDG in the GFP and Tat cell lines. (**B**) Tat binding at class I TDG blocks Pol II density at promoters. Box plots of normalized Pol II tag density at the promoter of class I (n = 11) or class II (n = 14) TDG in the GFP and Tat cell lines. (**C**) Normalized Pol II tag density relative to the Tat peak at class I TDG. (**D**) Normalized Pol II tag density relative to the Tat peak at class II TDG. (**E**) Pol II distribution at class I TDG (Metagene plots) in the Tat and GFP cell lines. (**F**) Genome browser views of ChIP-seq data at a class I TDG (*CD1E*) in the Tat and GFP cell lines. (**G**) Pol II distribution at class II TDG (Metagene plots) in the Tat and GFP. (**H**) Genome browser views of ChIP-seq data at a class II TDG (*EOMES*) in the Tat and GFP cell lines. This figure is associated with *Figure 7—figure supplement 1*. ChIP-seq, chromatin immunoprecipitation sequencing; GFP, green fluorescent protein; Pol, polymerase; TDG, Tat downregulated genes.
*Figure 7 continued on next page*

*Figure 7 continued*

The following figure supplement is available for figure 7:

**Figure supplement 1.** mTR reveals that Tat blocks both Pol II recruitment and pause release at TDG.

window (<10 Kb) from the Tat site, an insignificant number (~1%) contain a TAR-like motif in close proximity (<0.1 Kb) to the Tat site identified by ChIP-seq (*Figure 8C,D*). The data suggest that there is no significant presence of TAR-like motifs at or near Tat peaks, and only minimal examples of a TAR-like motif within 10 Kb of a Tat peak. Thus, it seems unlikely that the more prevalent mechanism of Tat recruitment to its target genes is through interaction with HIV TAR mimics.

Given that Tat's interaction with its target genes appears not to rely on TAR-like motifs, we reasoned that Tat might be recruited through interaction with master transcriptional regulators. To identify candidates, we submitted 200-bp windows surrounding Tat peaks from TSG and TDG to the DREME (Discriminative Regular Expression Motif) analysis tool (*Bailey, 2011*), which returned motifs matching T cell master regulators (ETS1, RUNX1) and GATA3 transcription factors for TSG, and ETS1 and RUNX1 for TDG, with statistically significant higher p-values in TSG ($2.5 \times 10^{-18}$ and $9.6 \times 10^{-14}$, respectively) (*Figure 8E,F*), without enrichment of other factors expressed in T cells such as members of the signal transducer and activator of Transcription (STAT) family. Remarkably, enrichment of both ETS1 and RUNX1 at TSG correlates with the observed functional annotation and their known role in modulating T cell activation (*Figure 1*) (*Hollenhorst et al., 2009*; *Hollenhorst et al., 2011*).

Given that ETS1 motifs are present at both TSG and TDG, we examined whether the presence of these motifs inform about the mode of Tat effect on cellular genes. To test this, we looked to see whether ETS1 motifs are more prevalent in class I or II TSG or TDG, and observed that the number of target genes containing ETS1 binding sites within 100-bp from the center of the Tat peak is: 11/17 (64.7%) for class I TSG, 36/43 (83.72%) for class II TSG, 8/14 (57.1%) for class I TDG, and 6/11 (54.5%) for class II TDG. This analysis indicates that the presence of an ETS1 motif at the Tat target genes is not by itself sufficient to determine the mechanism of action (stimulation or downregulation) or whether Tat modulates the initiation or elongation step of transcription. ETS1 may help recruit Tat to chromatin but it needs another determinant for specificity or mode of action, most likely related to the transcriptional activity status or yet unidentified co-factors (protein and/or long-non coding RNA), which might function in a gene-specific manner.

Given that the highest-confidence motif returned for both TSG and TDG is ETS1, we examined whether this was indicative of co-occupancy of ETS1 with Tat in these genes. We first retrieved ETS1 ChIP-seq data in Jurkat T cells (*Hollenhorst et al., 2009*) and compared locations of ETS1 peaks to the locations of Tat peaks in our dataset (*Figure 8G,H*). Contrary to our results when comparing TAR-like motifs sites to Tat ChIP-seq peaks, we discovered that 73% of TSG and 80% of TDG contain an ETS1 ChIP-seq peak within 100-bp of a Tat ChIP-seq peak. Together, the data suggest a model where Tat is recruited to host cell chromatin through interaction with T-cell identity factors such as ETS1 and not through TAR-like motifs, thus revealing unique and unexpected recruitment mechanisms. ETS1 ChIP-seq revealed 19,049 sites in the genome (*Hollenhorst et al., 2009*). Interestingly, nearly 52% of the Tat peaks detected by ChIP-seq (3203/6117) harbor an ETS1 binding event within 100-bp, significantly more than expected from random occurrence (p-value $1 \times 10^{-8536}$, Hypergeometric test).

Although a large number of both TSG and TDG contain ETS1 motifs, the enriched motifs are different between both classes (5'-A<u>GGAA</u>G/AT/C-3' and 5'-CN<u>GGAA</u>-3', respectively) (*Figure 8E,F*), even though both contain the signature motif for ETS1 binding (5'-GGAA-3'). Thus, it would be interesting to determine whether these different motifs dictate a diverse binding mode and could inform about the mechanism of activation and repression through the same transcription factor, and whether the DNA binding site directs recruitment of specific cofactors differentially targeted by Tat at TSG and TDG.

Given that we found no significant evidence of Tat interaction with TAR mimics in the human genome as well as evidence of Tat interaction with ETS1, we reasoned that one clear mechanism by which Tat is recruited to chromatin is through direct protein-protein interactions. To biochemically test that Tat is recruited to chromatin in a RNA-independent manner, we performed a cellular

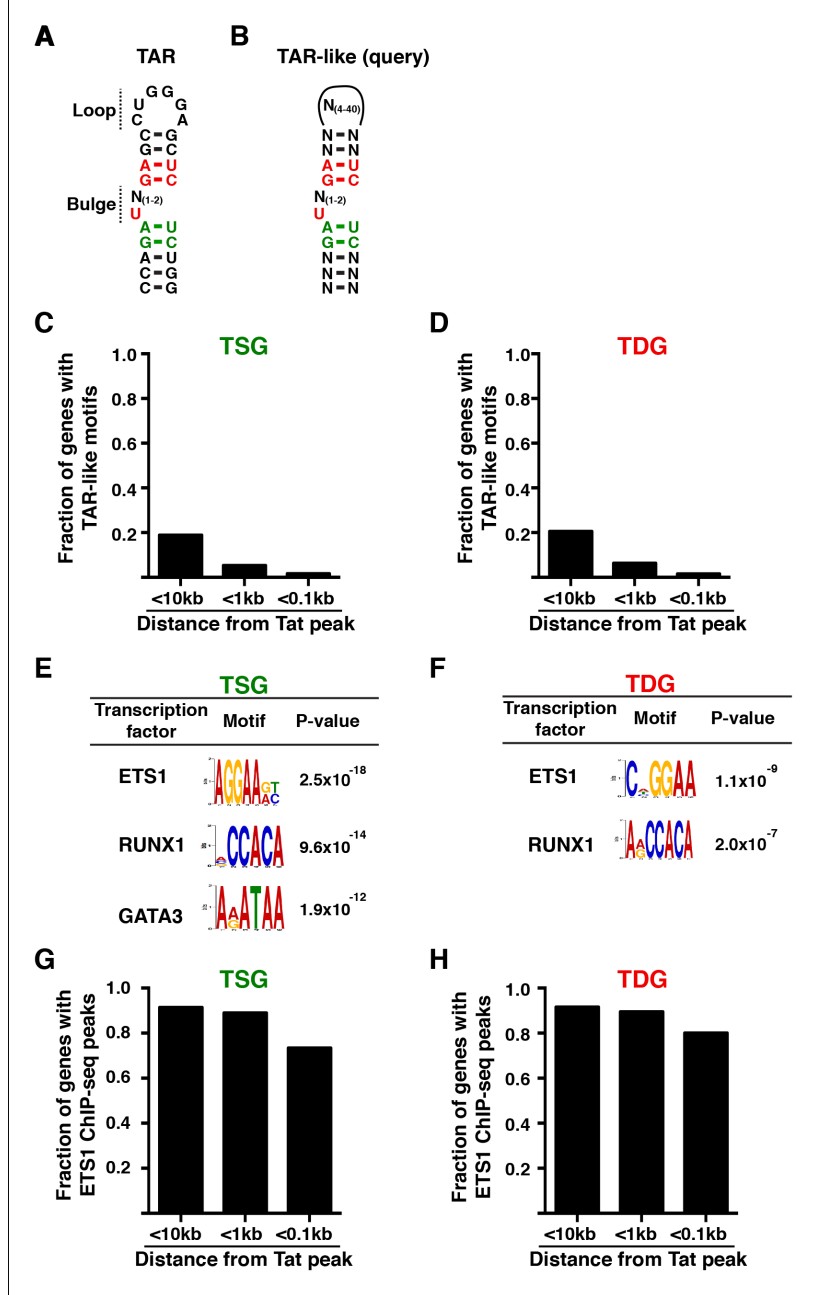

**Figure 8.** Enrichment of master transcriptional regulators, but not TAR-like, motifs on Tat sites at the direct target genes. (**A**) TAR sequence and scheme of the secondary structure. (**B**) Schematic of the sequence query used to search for TAR-like motifs within the direct Tat target genes. (**C**) TAR-like motifs are not found at or significantly near Tat peaks in TSG. Fraction of TSG containing TAR-like motifs at <10 kb, <1 kb or <0.1 kb from the Tat peak. (**D**) TAR-like motifs are not found at or significantly near Tat peaks in TDG. Fraction of TDG containing TAR-like motifs at <10 kb, <1 kb or <0.1 kb from the Tat peak. (**E**) MEME analysis of 200-bp windows surrounding Tat peaks in TSG reveals high-confidence motifs related to transcription factors ETS1 and RUNX1. (**F**) MEME analysis of 200-bp windows surrounding Tat peaks in TDG reveal different motifs for ETS1 and RUNX1. (**G**) Comparison of ETS1 ChIP-seq peak locations in Jurkat, as determined by Hollenhorst et al. (*Hollenhorst et al., 2009*), to Tat peak locations reveal that 73% of TSG contain an ETS1 peak within 100-bp of a Tat peak. (**H**) Comparison of ETS1 ChIP-seq peak locations to Tat peak locations reveal that 80% of TDG contain an ETS1 peak within 100-bp of a Tat peak. This figure is associated with *Figure 8—figure supplement 1*. ChIP-seq, chromatin immunoprecipitation sequencing; TAR, trans-activating RNA; TDG, Tat downregulated genes; TSG, Tat stimulated genes.

*Figure 8 continued on next page*

*Figure 8 continued*

The following figure supplement is available for figure 8:

**Figure supplement 1.** The interaction between Tat and host cell chromatin appears to be primarily dictated by protein–protein interactions.

fractionation by sequential salt extraction in the absence and presence of RNase (*Figure 8—figure supplement 1A*) and observed that: (i) Tat is present in the nucleoplasm (100 mM salt fraction) as well as bound to chromatin (eluted at 600 mM salt) from Jurkat nuclear extracts, while GFP is mainly detected in the 100 mM salt fraction; and (ii) the interaction between Tat and chromatin is primarily dictated by protein-protein interaction, although a minor fraction appears to be RNA-dependent (*Figure 8—figure supplement 1B*). To test this possibility further, we performed ChIP-qPCR assays at select TSG and TDG by preparing samples in the presence and absence of RNase and observed that the treatment does not affect Tat binding to four different target gene promoters (*Figure 8—figure supplement 1C*). Although we cannot strictly rule out that Tat can combinatorially interact with target proteins and RNA species present at different genomic domains, our data indicate that Tat interaction with chromatin is primarily dictated by protein-protein interactions.

## ETS1 recruits Tat to chromatin to reprogram cellular transcription

Given that we found a significant enrichment of ETS1 motifs in Tat sites (*Figure 8*) we reasoned that ETS1 mediates Tat recruitment to chromatin. To test whether the two proteins interact we performed Strep affinity purifications (AP) using nuclear fractions prepared from the Jurkat T-cell lines followed by western blot analysis. The data indicate that Tat, but not the C22A non-functional mutant or GFP, interacts with endogenous ETS1, as well as with the P-TEFb kinase (CDK9) used as positive control, thereby showing that the protein-protein interaction is specific (*Figure 9A* and *Figure 9—figure supplement 1A*). The facts that Tat and ETS1 interact, ETS1 motifs are enriched at the Tat target genes, and ETS1 binds these genes (as revealed by ETS1 ChIP-seq in Jurkat T cells [*Hollenhorst et al., 2009*]) prompted us to test whether both proteins co-occupy target genes. To test this possibility we performed ChIP assays on the Jurkat-GFP and -Tat cell lines using primer-pairs that amplify both promoter-proximal and promoter-distal regions of different Tat target genes. We found that Tat and ETS1 co-occupy the *CD69* promoter but not gene body (*Figure 9B*), consistent with the motif enrichment analysis and a previous ETS1 ChIP-seq dataset (*Hollenhorst et al., 2009*). Furthermore, Tat binding at the *CD69* promoter does not appear to alter the occupancy levels of ETS1, since the density of ETS1 in the absence and presence of Tat is similar, if not identical (*Figure 9B*). In addition, Tat binds some target genes lacking ETS1 motifs such as *CD1E* (*Figure 9C*). These results provide evidence that Tat also is recruited in an ETS1-independent manner thereby indicating that additional recruitment mechanisms exist.

Given that Tat and ETS1 interact and co-occupy target gene promoters, we asked whether ETS1 (already bound to DNA elements) recruits Tat to chromatin to reprogram gene transcription. To test this, we generated Jurkat-GFP and -Tat cell lines expressing ETS1-specific shRNAs to efficiently knockdown ETS1 or a non-target (NT) shRNA as negative control (*Figure 9D*). Remarkably, we observed that ETS1 knockdown consistently diminishes (~6-fold) ETS1 density at the *CD69* promoter in both GFP and Tat cell lines with a concomitant loss of Tat signal (~6-fold) at the promoter (*Figure 9E*). Importantly, ETS1 knockdown does not alter the recruitment of Tat to the *CD1E* promoter, which is regulated in an ETS1-independent manner (*Figure 9F*).

Given that Tat binds ETS1 and both co-occupy promoter-proximal regions of selected target genes, we examined whether ETS1 is critical for transcriptional activation of *CD69* and three other TSG and observed that ETS1 knockdown interferes with the Tat-mediated increase in target gene stimulation for both class I and II TSG (*Figure 9—figure supplement 1B–E*) without affecting RNA steady-state levels of non-target genes such as *RPL19* and *7SK* (*Figure 9—figure supplement 1F, and G*). The evidence that ETS1 is critical for Tat's transcriptional activation of these four selected TSG correlates with the Tat-mediated increase in the levels of Pol II and the chromatin marks coinciding with transcription initiation and elongation at both class I and II TSG (*Figures 2–5*).

Together, we provide compelling evidence that Tat is recruited to chromatin through interaction with the T-cell identity factor ETS1 to reprogram cellular transcription. Further studies are needed to

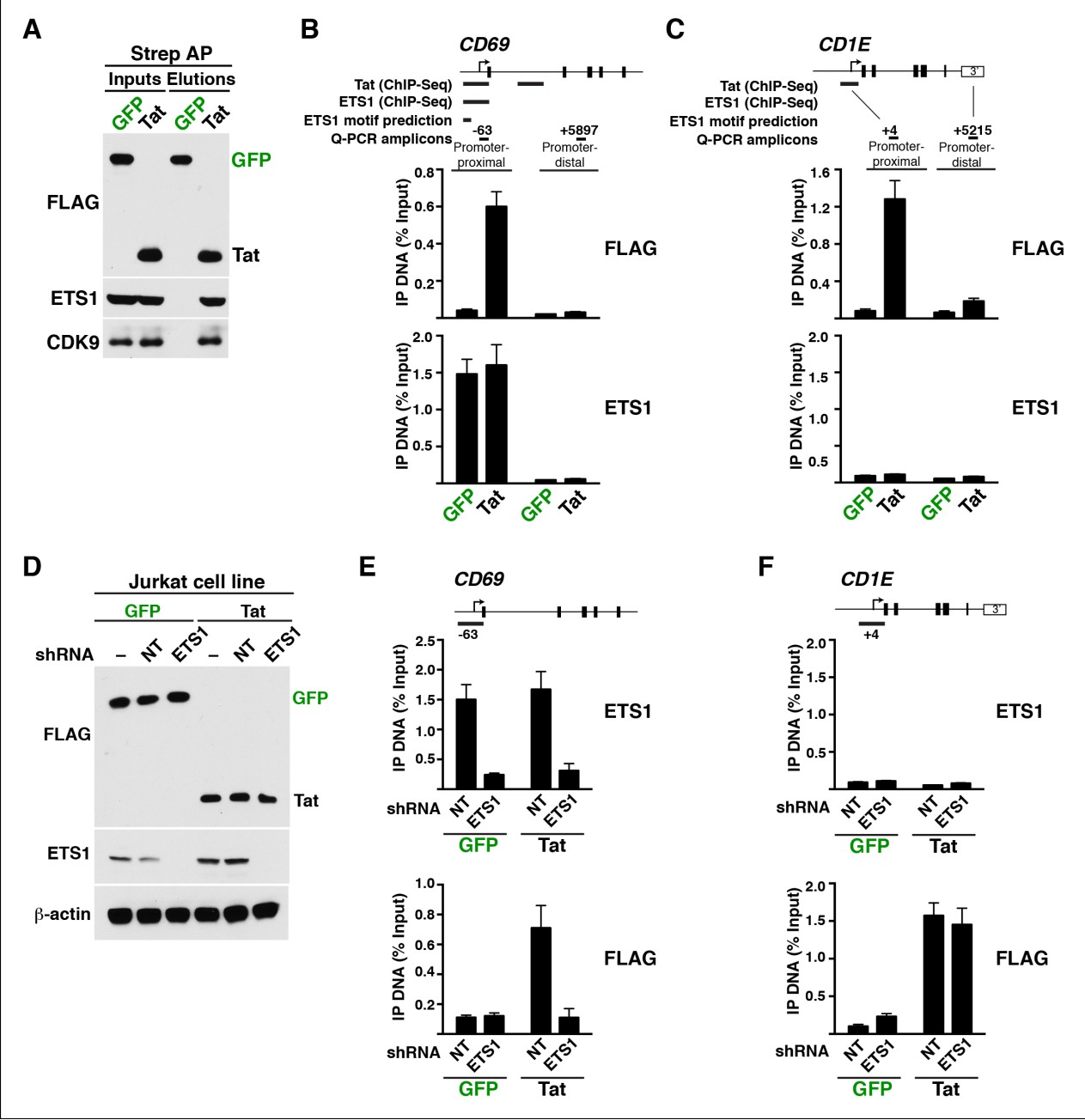

**Figure 9.** Tat is recruited to its target genes through interaction with the master transcriptional regulator ETS1. (**A**) Western blots showing interactions between Tat and ETS1. CDK9 was used as a positive control in the interaction. Strep-tagged Tat and GFP were AP from the Jurkat cell lines using Strep beads, and analyzed by western blot using the indicated antibodies. (**B**) Tat is recruited to target genes marked by ETS1. ChIP assays showing that Tat and ETS1 co-occupy the promoter-proximal but not promoter-distal region of *CD69*, in agreement with the location of Tat and ETS1 peaks found by ChIP-seq and the presence of ETS1 motifs (as revealed by enrichment analysis). (**C**) Tat is also recruited to target genes using ETS1-independent mechanisms. ChIP assays showing that Tat but not ETS1 occupies the promoter-proximal region of the *CD1E* target gene, in agreement with the Tat and ETS1 ChIP-seq dataset and motif prediction analysis. (**D**) Generation of Jurkat-GFP and -Tat cell lines expressing non-target (NT) or ETS1 shRNAs. (–) denotes the parental untransduced Jurkat cell lines. Protein lysates were analyzed by western blot using the indicated antibodies to verify for the RNAi efficiency. (**E**) ETS1 knockdown impairs Tat recruitment to the *CD69* promoter. ChIP-qPCR assays showing the density of ETS1 or FLAG at the *CD69* promoter in the Jurkat-GFP or -Tat cell lines transduced with the NT or ETS1 shRNAs. (**F**) ETS1 knockdown does not abolish Tat recruitment to the *CD1E* promoter. ChIP-qPCR assays showing the density of ETS1 or FLAG at the *CD1E* promoter in the Jurkat-GFP or -Tat cell lines transduced with the NT or ETS1 shRNAs. This figure is associated with *Figure 9—figure supplement 1*. AP, affinity purified; ChIP-seq, chromatin immunoprecipitation sequencing; GFP, green fluorescent protein; NT, non-target; qPCR, quantitative polymerase chain reaction; shRNA, small hairpin RNA.

*Figure 9 continued on next page*

*Figure 9 continued*

The following figure supplement is available for figure 9:

**Figure supplement 1.** The C22A non-functional Tat mutant fails to bind ETS1 and evidence that ETS1 is critical for transcription activation of Tat target genes.

determine whether ETS1 functions as a scaffold to promote Tat recruitment or whether Tat induces protein conformational changes to activate or repress ETS1 regulatory transcriptional programs (see Discussion).

## Discussion

Several studies have reported that Tat binds the human genome to modulate cellular gene expression to alter the biology of immune cells (dendritic and CD4+ T) and generate a permissive environment for viral replication and/or spread (*Huang et al., 1998*; *Izmailova et al., 2003*; *Kim et al., 2010a*; *2013*; *Kukkonen et al., 2014*; *Li et al., 1997*; *Marban et al., 2011*). However, a comprehensive description of the direct target genes and the nature of the regulatory mechanisms have yet to be discovered. In this article, we report a technically improved Tat ChIP-seq assay that dramatically increases sensitivity compared with previous methodologies. Furthermore, by simultaneously assaying transcriptome changes and Tat's genome-wide distribution we combinatorially identified direct Tat targets in the human genome with high confidence. In contrast to previous studies, we were able to elucidate a large proportion of Tat binding sites in the human genome that correlate with marked gene expression changes (activation or repression) at both the RNA and protein levels. Importantly, the Tat target genes shared functional annotations, and are regulated (for the most part) by a common set of master transcriptional regulators.

Transcription factors coordinate the activation and maintenance of transcriptional programs by regulating one or multiple steps in the transcriptional cycle. While some sequence-specific DNA binding transcription factors recruit Pol II and the basal transcription apparatus to promote transcription initiation (*Hahn, 2004*), others function at the elongation step by allowing Pol II transition from a promoter paused state to the productive elongation phase (*Adelman and Lis, 2012*; *Rahl et al., 2010*; *Zhou et al., 2012*). By performing a global analysis of chromatin signatures that are generally associated with different genomic domains (promoter, coding units and enhancers) together with genome-wide Pol II distribution data in the absence and presence of Tat, we described, for the first time, the precise nature of the mechanisms of transcriptional activation and repression by Tat. Strikingly, we found that Tat directly controls both the initiation and elongation steps to transcriptionally reprogram the cell. Tat promotes Pol II recruitment at promoters and transcribing units to stimulate the transcription initiation and elongation steps, respectively. Conversely, Tat blocks Pol II recruitment to promoters or transcribing units to prevent the initiation and elongation steps, respectively, thereby leading to gene repression. Remarkably, the global analysis of chromatin signatures is consistent (for the most part) with the proposed mechanisms based on Pol II occupancy changes. For example, class I TSG (regulated at the initiation step) showed a large increase in both Pol II and H3K4me3 at promoter-proximal regions, consistent with their functional link in activating gene transcription (*Shilatifard, 2012*). Although we have not searched for the H3K4 methylase, it would be interesting to define whether Tat hijacks one or more of Set/MLL methylases and whether DoT1L and the SEC are co-recruited to link transcription initiation with elongation (*Luo et al., 2012*; *Nguyen and Zhang, 2011*; *Shilatifard, 2012*). In addition, the mechanism by which Tat reduces H3K4me3 density at some class II TSG remains yet unknown, but it may be possible that Tat recruitment to the promoter region interferes with the H3K4me3 signature, as has been seen with the NS1 protein during influenza virus infection (*Marazzi et al., 2012*). Although we have described that Tat associates with and recruits chromatin-modifying enzymes to target genes, one important aspect of the mechanism that we have not clarified yet is that the histone modifications may represent (in some cases) indirect effects of changes in transcription and not a direct consequence of Tat function on the genome.

Despite the canonical view of transcriptional regulation through transcription factor-promoter DNA interaction, it has recently become evident that internal promoters, intragenic enhancers and

other genomic elements contribute to specify transcriptional programs through the formation of three-dimensional structures (*Bulger and Groudine, 2011*; *Creyghton et al., 2010*; *Ghavi-Helm et al., 2014*; *Heintzman et al., 2007*; *Jin et al., 2013*). Consistent with these discoveries, we found that Tat exploits the human genome by binding not only at promoters but also at intergenic and intragenic sites to modulate long-range chromatin interactions and transcription activation from these promoter-distal sites. We observed that at a large number of class I TSG (regulated at the initiation step) Tat binds at intragenic sites to modulate enhancer activity. In the absence of Tat, the nucleosomes surrounding these sites contain low or undetectable histone modifications related to enhancer activity (H3K4me1 and H3K27Ac). Tat binding increases their density in a manner that is proportional to transcriptional levels. Notably, this increase also correlates with the recruitment of chromatin-modifying enzymes (p300/CBP) at the Tat site, chromatin looping between the internal site and the promoter, and Pol II recruitment to trigger de novo transcription initiation. The role of intragenic Pol II pausing requires further investigation but it may be possible that this form of Pol II is required for active transcription elongation by promoting template DNA circularization or acting after gene looping. This Pol II form might not have been observed in embryonic stem cells because most, if not all, genes are incompetent for elongation and Pol II is primarily paused in the promoter-proximal region.

Recent evidence suggested that internal sites marked with H3K27Ac appeared to be a sort of intragenic enhancers (*Kowalczyk et al., 2012*), implying that Tat directly dictates gene activation by binding at these sites and is likely to be a major determinant of the overall architecture and the composition of histone modifications of these internal sites. The gene looping hypothesis provides a mechanistic explanation for the molecular effects observed at the promoter (namely Pol II recruitment and increase in H3K4me3 density) in response to Tat binding at promoter-distal sites, even without detectable Tat at the promoter-proximal region. Although we have shown that the C22A non-functional mutant (which does not dimerize) does not promote gene looping, it would be interesting to further test whether the Tat-mediated long-range chromatin interactions are controlled by protein homo-dimerization, a feature of Tat that was originally described by Frankel and Pabo (*Frankel et al., 1988*), but whose function has since then remained largely elusive. Furthermore, the role of intragenic enhancers in activating gene expression is an attractive possibility, and might help recruit chromatin-modifying enzymes for compartmentalization purposes and/or 'on site' activation.

Although in this report we did not thoroughly examine Tat binding at intergenic regions and their functional role in transcriptional control, it is plausible that Tat binds these genomic domains (enhancers) to control long-range interactions and the assembly of transcription complexes to modulate transcription activation/repression. Given that transcription from enhancers is a widespread regulatory mechanism to modulate the activity of nearby genes, further research is needed to understand the role of Tat in controlling transcription through enhancers and the potential role of the newly identified class of enhancer-derived non-coding RNAs (eRNAs) in transcription activation or repression (*Kim et al., 2010b*; *Kim and Shiekhattar, 2015*). Although we have detected Tat-induced RNAs from enhancers that are co-regulated with nearby genes, further investigation is also needed to define the molecular mechanisms by which these Tat-induced non-coding RNAs function during the reprogramming process.

Although the majority of transcription factors described to date contact DNA directly, Tat is unique because it binds the nascent RNA structure (TAR) formed at the HIV promoter. Surprisingly, in contrast to activation of the HIV genome, we found insignificant enrichment of TAR-like motifs at Tat binding sites in the human genome. Although we cannot completely exclude the possibility that nascent RNA chains (folding or not into TAR-like structures) help Tat recruitment to host cell chromatin, we unexpectedly found that Tat occupies sites bound by master transcriptional regulators (ETS1) with high frequency. Notably, we provided biochemical and genetic evidence that ETS1 recruits Tat to chromatin to modulate (activate or repress) gene transcription. ETS1 is a member of a large family of transcription factors (ETS) that play important roles in T-cell stimulation and differentiation (*Hollenhorst et al., 2011*). ETS1 requires combinatorial interactions with other factors (such as RUNX1) to activate transcription. We propose that, similarly to RUNX1, Tat uses ETS1 as a scaffold to promote its recruitment to target genes and modulate gene transcription. Given that ETS1 is found at both activated and repressed genes, it is evident that ETS1 does not dictate per se the mode of regulation (activation or repression). For the mechanism of gene activation, a likely scenario would be that Tat is recruited to ETS1 to relieve its auto-inhibition and recruit Pol II, elongation

factors and chromatin-modifying enzymes. For the mechanism of gene repression, Tat might compete off factors pre-associated with ETS1 (such as RUNX1) and block PIC assembly (in the case of transcription initiation blockage) or prevent the action of elongation factors such as P-TEFb (in the case of transcription elongation repression). Although ETS1 appears to play a central role in the recruitment of Tat to several direct target genes (irrespective of the transcriptional outcome), it is completely possible that several other co-factors including long non-coding RNAs and/or proteins function combinatorially to specify target loci identification and Tat function.

Although we have provided important insights into the recruitment mechanisms, further investigation is needed to clarify the molecular basis of the Tat-ETS1 protein-protein interaction in the regulation of gene activation and repression. One particular observation derived from the motif analysis is that the ETS1 motif identified at both TSG and TDG contain a common sequence (5'-GGAA-3') but differ in the -1/-2 positions. Of note, transcription factor binding sites slightly differing in sequence have been shown to modulate various steps of the transcription cycle including transcription factor binding affinity and conformational changes upon their recruitment to target sequences, as well as co-factor recruitment (*Meijsing et al., 2009*). Therefore, the difference in the ETS1 motifs at TSG and TDG might contain information utilized by Tat (and/or Tat co-factors) to activate or repress transcription. Undoubtedly, further research is required to elucidate the precise molecular basis.

Despite the widespread role of ETS1 in recruiting Tat to chromatin, it is most likely that other recruitment strategies exist because not all Tat target genes identified are regulated by ETS1. Defining all the details will require higher-resolution approaches such as ChIP-exo and ChIP-nexus to improve the definition of Tat target sites and mechanisms of recruitment to chromatin (*He et al., 2015*; *Rhee and Pugh, 2011*). In addition, the analysis of transcriptome changes using RNA-seq (which measures steady-state RNA levels and not transcription per se) has some caveats, for example difficulties in detecting low abundance or highly unstable RNAs. Potentially, this may help explain why we also found a large fraction of binding events that do not have a correlate with gene expression changes. Other tools such as GRO-seq (*Core et al., 2008*), which measure levels of nascent RNAs, will also be needed to further improve the definition of direct and indirect target genes. Moreover, a more recent and powerful technique named NET-seq yields transcriptional activity at nucleotide resolution and thus outperforms GRO-seq (*Mayer et al., 2015*). Certainly, these tools will provide much higher-resolution to define target sites, modes of Tat recruitment and improved functional insights. Nonetheless, our work provides the first comprehensive view of how Tat modulates the biology of immune target cells by mediating key transcriptional changes. Interestingly, Tat can potentially perform different functions depending on the target cell type. Thus, it would be informative to determine how host adaptability is harnessed by a viral protein to selectively reprogram transcription in a cell lineage-specific manner.

In conclusion, despite the complexity of transcriptional regulatory mechanisms in the cell, Tat precisely controls Pol II recruitment and pause release to fine-tune the initiation and elongation steps, respectively. It is possible that the diversity of mechanisms employed by Tat to reprogram the host cell arise from the intrinsic complexity of transcriptional regulatory strategies of human genes. Finally, our data provide yet another example on how a virus with a limited coding capacity optimized its genome by evolving a small but 'multi-tasking' protein to simultaneously control viral and cellular transcription using distinct regulatory strategies.

## Materials and methods

### Experimental analysis

#### Cell culture

HEK293T cells were maintained in DMEM with 10% fetal bovine serum (FBS), 100 U/ml penicillin and 100 µg/ml streptomycin (Life Technologies, Foster City, CA). CD4+ Jurkat T cells (clone E6.1) were cultured in RPMI 1640 (HyClone) with 10% FBS, 100 U/ml penicillin and 100 µg/ml streptomycin (Life Technologies). The Jurkat HIV E4 clone was kindly provided by Dr. J. Karn and described elsewhere (*Pearson et al., 2008*). Jurkat T-REx cells were maintained in the same conditions as for Jurkat but in the presence of 10 µg/ml Blasticidin as indicated by the manufacturer's instructions (Life

Technologies). The derived Jurkat T-REx clones (GFP and Tat bearing the Strep/Flag (SF) epitope: GFP-SF and Tat-SF, respectively) were selected with 300 µg/ml of zeocin for four weeks and later cultured with 10 µg/ml Blasticidin and 100 µg/ml zeocin (Life Technologies). The Tat cell line was cultured for less than 3 months because it otherwise loses Tat expression and responsiveness. To generate the stable GFP:SF and Tat:SF expressing cell lines, the parental Jurkat T-REx was electroporated with 2 µg of Ssp1-linearized pcDNA4/TO vector (Life Technologies) bearing either insert using a nucleofector kit V and a Nucleofector II electroporator (Lonza, Walkersville, MD). To induce expression of GFP:SF and Tat:SF, the stable cell lines were treated with 1 µg/ml doxycycline (DOX, Sigma, St. Louis, MO) for 16 hr (or as indicated). We chose to utilize a SF-tagged Tat protein and immunoprecipitate Tat:SF using an anti-FLAG antibody due to our inability to efficiently immunoprecipitate Tat under denaturing conditions with the available commercial anti-Tat antibodies from Covance (catalogue number MMS-116) and Abcam (ab43014).

## Primary CD4+ T cell isolation, stimulation and HIV infection

Blood from healthy donors was obtained from the Gulf Coast Regional Blood Center from Houston, TX. Peripheral blood mononuclear cells, were isolated by Ficoll-Hypaque density gradient centrifugation (Stemcell Technologies, Vancouver, BC, Canada) and naïve CD4+ T cells were then purified by negative selection using the magnetic beads human-naïve CD4 T cell enrichment set (Becton Dickinson, 558521). Cell purity was determined by surface staining: APC-conjugated anti-CD4 (555349, Becton Dickinson, Franklin Lakes, NJ) and fluorescein isothiocyanate-conjugated CD45RA (555488, Becton Dickinson). To generate CD4 central memory T cells ($T_{CM}$), naïve CD4 T cells were activated at day 0 (*Figure 1—figure supplement 8*) in 96-well plates precoated with anti-CD3/CD28 [2.5 µg/ml anti-CD3 (OKT3), 2.5 µg/ml anti-CD28 (Biolegend, 302914)] in the presence of anti-IL-12 (4 µg/1 × $10^6$ cells; R&D, MAB219), anti-IL-4 (2 µg/1 × $10^6$ cells; MAB204, R&D, Minneapolis, MN ) and TGF-β1 (0.8 µg/1 × $10^6$ cells; 100-21, Peprotech, Rocky Hill, NJ) (*Bosque and Planelles, 2011*; *Messi et al., 2003*). After activation cells were maintained at 1 × $10^6$ cells/ml in RPMI 1640 with L-Glutamine (HyClone) containing 10% FBS, 100 U/ml penicillin and 100 µg/ml streptomycin (Life Technologies) in the presence of 30 IU/ml of human IL-2 (AIDS Research and Reference Reagent Program, Germantown, MD). For infection of $T_{CM}$ with replication competent X4 virus (pNL-4-3–GFP Nef (–), MOI = 0.5), activated cells (day 7) were spinoculated (1 × $10^6$ cells in 1 ml; 2 hr, 2900 rpm) in the presence of 8 µg/ml polybrene (hexadimethrine bromide, Sigma). After infection, the supernatant was removed, cells were resuspended in complete RPMI and crowded for 3 days in 96-well plates (round bottom) at a concentration of 1 × $10^6$ cells/ml (0.1 ml/well). Three days post-infection GFP+ cells were sorted at the UTSW Flow Cytometry Core. At this point, the supernatant and cells were harvested for p24 enzyme-linked immunosorbent assay (ELISA) and RNA isolation, respectively, as indicated above.

## Protein–protein interaction assays

Tat-associated proteins were purified using Strep-Tactin Superflow beads (IBA Life Sciences, Olivette, MO). Briefly, Jurkat (1 × $10^9$ cells) were lysed using the Dignam method with brief modifications (*Dignam et al., 1983*). Strep-Tactin beads were equilibrated with IP lysis buffer, mixed with cell lysates and rotated at 4°C for 2 hr. After centrifugation at 3000 rpm for 1 min, unbound proteins were removed and beads washed four times with IP wash buffer (20 mM Tris-HCl pH = 7.5, 1.5 mM $MgCl_2$, 250 mM NaCl, 0.2% NP-40, 5% glycerol, 1 mM DTT, protease inhibitor [Roche, Pleasanton, CA]). Specifically bound proteins were eluted with elution buffer containing desthiobiotin (IBA Life Sciences) for 30 min at 4°C.

## Cell fractionation by sequential salt extraction

1 × $10^8$ cells (Jurkat-GFP and -Tat) were collected from 100 ml of culture media (grown at 1 × $10^6$ cells/ml) and washed with 2 ml of cold 1× phosphate buffered saline (PBS) plus ethylendiaminetetraacetic acid (EDTA)-free protease inhibitor (PI) cocktail (Roche). The cell pellet (PCV = 0.1 ml) was resuspended in 1 ml buffer A (10 mM KCl, 10 mM HEPES pH 7.9, 0.1 mM EDTA, 1 mM DTT, 0.4% NP-40, plus PI) and incubated on ice for 5 min. Nuclei were pelleted by centrifugation (5 min spin at 2,000 g at 4°C), the supernatant saved as the cytoplasmic extract (~0.5–0.6 ml), and the nuclear pellet was washed twice, with 1 ml buffer A. Nuclei were resuspended in 0.25 ml

buffer B (20 mM HEPES pH 7.9, 100 mM NaCl, 1 mM EDTA, 1 mM DTT, 1% NP-40 plus PI) using 10 strokes in a 1-ml dounce homogenizer and centrifuged for 5 min at 6000 g. The supernatant (~0.4 ml) was designated as nuclear extract (100 mM). The pellet was similarly resuspended in 2× pellet volume of buffer B containing 600 mM NaCl, vortexed for 10 s, treated with RNAse A (Roche) or BSA (mock treatment) at a concentration of 10 µg/ml for 60 min at 4°C and then spun at 6000 g for 5 min.

## Western blotting and antibodies

Protein samples were resolved in sodium dodecyl sufate (SDS)-polyacrylamide gel electrophoresis, transferred to 0.45 µm nitrocellulose (Bio-Rad, Hercules, CA) membranes, blocked in Tris-buffered saline (TBS) containing 5% non-fat dry milk for 1 hr, and incubated with primary antibodies overnight at 4°C. Primary antibodies used for western blot: FLAG M2 (F1804, Sigma); STREP-Tag II (71591, Novagen, Madison, WI); CDK9 (sc-484, Santa Cruz Biotechnologies, Dallas, TX); ETS1 (sc-350X, Santa Cruz Biotechnologies); β-actin (sc-47778, Santa Cruz Biotechnologies); Tat (ab43014, Abcam, Cambridge, MA); Dot1L (A300-954A, Bethyl, Montgomery, TX); SetD2 (ab69836, Abcam); histone H3 (ab1791, Abcam). Secondary antibodies coupled to horseradish peroxidase (HRP) were donkey anti-rabbit IgG-HRP (sc-2313, Santa Cruz Biotechnologies), goat anti-mouse IgG-HRP (sc-2005, Santa Cruz Biotechnologies), and donkey anti-goat IgG-HRP (sc-2020, Santa Cruz Biotechnologies) and were incubated at 1:10,000 dilutions for 1 hr and blots developed using Clarity Western ECL Substrate kit (Bio-Rad).

## Flow cytometry

For cell surface marker staining, Jurkat cells were washed twice with PBS/0.5% BSA, and incubated with 50 µl of the antibody diluted in PBS/0.5% BSA for 30 min in the dark. Antibodies used include CD69-PE (12–0699, eBioscience, San Diego, CA) or Mouse IgG1 K isotype control PE (12–4714, eBioscience) and CD4-APC (MHCD0405, Caltag, Buckingham, UK). Cells were washed twice with PBS and then resuspended in 200 µl PBS/2.5% formalin, washed twice with PBS and twice with PBS/0.5% BSA. Fixed cells were subjected to flow cytometry analysis on a FACS Calibur in the Flow Cytometry Core Facility in UT Southwestern Medical Center. Data analysis was performed using FlowJo version 9.6.1.

## Enzyme-linked immunosorbent assay (ELISA)

Viral stocks and kinetics of HIV p24 antigen expression were monitored by ELISA following the manufacturer's instructions. Plates and reagents were obtained from SAIC Frederick, Inc. from the National Cancer Institute's (NCI) Operations and Technical Support Contractor to the Frederick National Laboratory for Cancer Research (FNLCR).

## RNAi

Lentiviral particles were made by transfecting HEK293T cells with three plasmids: the shRNA transfer vector pLKO.1 (Sigma), pMD2. G (VSV-g) and psPAX2 (gag-pol). Supernatants were harvested 2 days after transfection, quantified by ELISA and stored at –80°C. Jurkat cells ($2 \times 10^5$) were spinoculated at 2900 rpm for 2 hr with lentiviral particles (50–200 µl, ~$1 \times 10^7$ transducing units determined by p24 ELISA) in the presence of 8 µg/ml polybrene. Efficiently transduced cells were selected with 1 µg/ml puromycin for 5 days. At this point, cells were used for validation of RNAi efficiency by qRT-PCR and/or western blot using the corresponding primer-pair and primary antibody, respectively. shRNAs used for RNAi are: pLKO.1-NT shRNA (SHC002, Sigma); pLKO.1-Dot1L shRNA (TRCN0000020209, Sigma); pLKO.1-SetD2 shRNA (TRCN0000003030, Sigma); pLKO.1-CDK9 shRNA (TRCN0000000494, Sigma); pLKO-puro-IPTG-3×LacO-NT shRNA (SHC332, Sigma); pLKO-puro-IPTG-3×LacO-ETS1 shRNA (TRCN0000005591, Sigma).

## Chromosome conformation capture assay

Chromatin interaction was determined using a 3C assay (*Hagege et al., 2007*). For the 3C assay in the absence/presence of CDK9 inhibitor, cells were induced with DOX for 6 hr (minimal detectable time before the onset of protein induction) followed by 2 h incubation with FP or DMSO (vehicle), and cells were then processed as follows. $1 \times 10^7$ Jurkat cells were fixed with 1.5% methanol-free

formaldehyde (Thermo Fisher, Waltham, MA) at room temperature for 10 min, the crosslinking was quenched with 0.125 M glycine and then cells washed twice with cold PBS. Cell pellets were homogenized in cold lysis buffer (10 mM Tris-HCl pH 8.0, 10 mM NaCl, 0.2% NP-40) using a dounce homogeneizer (10 cycles) and incubated for 90 min at 4°C with constant rotation. After centrifugation for 5 min at 400 g at 4°C, $1 \times 10^6$ nuclei were re-suspended in 0.5 ml of $1.2 \times$ restriction enzyme buffer for DNA digestion followed by incubation with 7.5 µl of 20% (w/v) SDS for 1 hr at 37°C with constant shaking. 50 µl of 20% (v/v) Triton X-100 were treated for 1 hr at 37°C with constant shaking. Chromatin DNA was digested with 600 units of the indicated restriction enzyme (EcoRI) overnight at 37°C with constant shaking. Samples were incubated with 40 µl of 20% (w/v) SDS for 30 min at 65°C to stop the reaction. Digested nuclei were transferred to 50 ml Falcon tube and incubated with 6.125 ml of 1.15 X ligation buffer and 375 µl of 20% (v/v) Triton X-100 for 1 hr at 37°C with constant shaking. DNA ligation was performed by incubation with 2000 units of T4 DNA ligase (New England Biolabs) for 4 hr at 16°C followed by 30 min at room temperature. Reverse crosslinking was done by incubation with 25 µl (20 mg/ml) of proteinase K (Roche) at 65°C overnight. DNA samples were then treated with 5 µl of RNase A (Roche) for 30 min at 37°C, and then mixed with 7 ml of phenol–chloroform vigorously and centrifuged for 15 min at 2200 g at room temperature. The supernatant was transferred into a 50 ml tube and mixed with 7 ml of distilled water, 1.5 ml of 2 M sodium acetate pH 5.6, and 35 ml of ethanol to precipitate DNA by centrifuging at 12,000 g for 30 min at 4°C. The DNA pellet was dissolved in 150 µl of 10 mM Tris-HCl pH 7.5. Random ligation matrix was prepared by digestion with restriction enzyme and ligation with BAC clones containing the respective genes examined from the Children's Hospital Oakland Research Institute (CHORI, Oakland, CA). DNA concentration of 3C samples was determined by qRT-PCR assays with GAPDH-specific primers and using Fast SYBR Green Master Mix in a 7500 Real-Time PCR System (Applied Biosystems, Carlsbad, CA). Primers used for 3C will be provided upon request.

## RNA extraction and qRT-PCR assays

Total RNA from the Jurkat cell lines was isolated using TRIzol (Life Technologies) and RNeasy mini kit (Qiagen, Valencia, CA). 1 µg of total RNA was used for synthesis of first strand cDNA with M-MLV (New England Biolabs, Ipswich, MA), and qPCR was performed using a Fast SYBR Green Master Mix in a 7500 Fast Real-Time PCR System (Applied Biosystems). Experiments were done in biological triplicates and error bars represent the SEM as indicated in all figure legends. The primers used for qRT-PCR analysis are as follow (Gene, forward primer, reverse primer):

ACTB (CCCCCCGGGCCGTCTTCCCCTC, TGAGGATGCCTCTCTTGCTCTG)
RPL19 (ATCGATCGCCACATGTATCA, GCGTGCTTCCTTGGTCTTAG)
7SK (TAAGAGCTCGGATGTGAGGGCGATCTG, GGAGCGGTGAGGGAGGAAG)
CD69 (ACTGTGAAGAGGAGCTGG, GTGTTCCTCTCTACCTGCGTATCG)
ADCYAP1 (GGGATCTTCACGGACAGCTA, CGGCGTCCTTTGTTTTTAAC)
FAM46C (GCCTAGGGGCTGTAGAGGTCG, CTGCTCTCCTCTGCCATCTT)
PPM1H (GCACACACAATGAAGACCAAGCCA, CAATAGTGGCAGGAAACACCCTCG)
ANXA1 (ACGCTTTGCTTTCTCTTGCTAAGG, AAGGCCCTGGCATCTGAATCAGCC)
ZNF83 (AGCCACCAAGAAGACCAAAGAAA, TATAAAGCCCTCTGTGCAGGGTTCA)
CD1E (AAAGCCTTCTTGGTCACACCT, GCTTTGGGTAGAATCCTGAGAC)
FBLN2 (GCCGATGGCTATATCCTCAAT, CAGGTGAGTGCCTTGTAGCA)
EOMES (AAATTCCACCGCCACCAAACTGAG, TTGTAGTGGGCAGTGGGATTGAGT)
CDK6 (ATGTGTGCACAGTGTCACGAACAG, TTAGATCGCGATGCACTACTCGGT)
PRAME (ATTTGCGGAGGTTTTCACAGGA, TATCGGCTCTGAATGGAACCCC)
FAM133B (TCGGGTGGCCTATATGAACCCAAT, GCCAAAGCCTTGGAGCCTTTCTTT)
Dot1L (CCACCAACTGCAAACATCACT, GAGGAAATCGCCTCTCTCCAAT)
SetD2 (CTCTCACCACCCTCTTCTGCCTA, CCACCTCTTGCTCAAACAACTTCC)
CDK9 (GTGTTCGACTTCTGCGAGCATGAC, CTATGCAGGATCTTGTTTCTGTGG)
PPM1H eRNA (AGCCTGGAGTGGAGAAGAAA, TCTGATTCCCTTCCATCCTC)
GAPDH (CCCTGTGCTCAACCAGT, CTCACCTTGACACAAGCC)
CD82 (GGGCTCAGCCTGTATCAAAG, CAGGACAGAGATGAAACTGCTC)
VAV3 (CAACATCCTCCCTCCTTAGC, CCCTGACAATGACAGACTGG)
HSPA8 (CAGTACGGAGGCGTCTTACA, CCACTTGGGTGGAGAAGATT)

CD69-In (TCTTGTCCACTCTCCGGATGC, AGACTCAACAAGAGCTCCAGC)
PPM1H-In (CTCCTTGAGGATGAGGATGG, CTCAGGACGAGGTGGAGTG)
VAV3-In (GCAGAGCAGGACTCCATCGCGG, AGCCGCGTCCCGGAGCCGTCG)
ANXA1-In (CAAACAGAAGGCAGCCAAT, CAGTGGAGACTTGGGCTTCT)
CD1E-In (CAGTTAGTGCAATTAGGGAG, AAGCCACACTGCCTCGC)
FBLN2-In (CGCGCACACAGCCAGGGG, GCGGACCGACGGGGAT)
EOMES-In (ACGCTGGAAGAAGGTGACTT, TCAACTTGACCGATGCTTTG)
CDK6-In (AGAGTTCCAGTTCCGTTTGG, GATCCTGCTTCCACGTATCC)

## RNA-seq library preparation

For RNA-seq, total RNA samples (10 µg) were prepared from biological duplicate samples with TRI-zol followed by depletion of rRNA using the Ribominus isolation kit (Life Technologies). Briefly, 500 ng of rRNA-depleted RNA were used for library preparation using the SOLiD Total RNA-seq kit (Applied Biosystems). The RNA was fragmented and adapters ligated before cDNA synthesis. The cDNA was then size selected, amplified, purified with AmpureXP beads (Beckman Coulter, Brea, CA) and quality checked on a Bioanalyzer (Agilent, Santa Clara, CA). Samples were quantified by qPCR using SOLiD adapter primers per the manufacturer's instructions (Applied Biosystems) and the EZ-Bead system was used to amplify the libraries onto the sequencing beads for high-throughput sequencing on a SOLiD 5500xl instrument (Applied Biosystems).

## Chromatin immunoprecipitation assay

Chromatin was cross-linked with 1% formaldehyde for 10 min at room temperature in culture media, and the reaction was stopped by the addition of glycine (125 mM final). Cells were washed twice with PBS and re-suspended in 1 ml of nuclei extraction buffer (5 mM PIPES pH 8.0, 85 mM KCl, 0.5% NP-40, 1 mM PMSF, protease inhibitor [Roche]) at $2 \times 10^7$ cells/ml and incubated for 10 min at 4°C. Cell nuclei were collected by centrifugation at 3000 rpm for 5 min at 4°C, and re-suspended in Szak's RIPA buffer (50 mM Tris-HCl pH 8.0, 1% NP-40, 150 mM NaCl, 0.5% deoxycholate, 0.1% SDS, 5 mM EDTA, 0.5 mM PMSF, and EDTA-free protease inhibitor cocktail [Roche]) at $5 \times 10^7$ nuclei/ml. Nuclear pellets were sonicated using the Bioruptor water bath (Diagenode) using the high setting with 30 sec on 30 sec off for 45 min to generate 100–300 bp DNA fragments. Chromatin DNA was quantified with NanoDrop 1000 (Agilent). For the ChIP step, 1–3 µg of antibody was conjugated with 15 µl of protein G dynabeads (Life Technologies) and blocked with 0.16% bovine serum albumin for 16 hr at 4°C. Antibody-conjugated dynabeads were incubated with 30–50 µg of chromatin DNA for 2 hr at 4°C. Then the beads were sequentially washed two times with the following buffers: low salt buffer (20 mM Tris-HCl pH 8.0, 150 mM NaCl, 1% Triton, 0.1% SDS, 2 mM EDTA), high salt buffer (20 mM Tris-HCl pH 8.0, 500 mM NaCl, 1% Triton, 0.1% SDS, 2 mM EDTA), LiCl wash buffer (0.25M LiCl, 1% NP-40, 1% deoxycholate, 1 mM EDTA, 20 mM Tris-HCl, pH 8.0), and 1 mM Tris-EDTA pH 8.0. Chromatin immunocomplexes were eluted by incubation for 10 min at 65°C with 1% SDS and 100 mM NaHCO$_3$, and cross-linking was reversed by incubation in the solution adjusted to 200 mM NaCl and proteinase K (20 µg) for 1 hr at 65°C. Antibodies used for ChIP: RNA polymerase II unphosphorylated/Ser5P-CTD (39097, Active Motif); RNA polymerase II total P-CTD (61081, Active Motif); RNA polymerase II Ser5P-CTD (ab5131, Abcam); RNA polymerase II Ser2P-CTD (ab5095; Abcam); Normal rabbit IgG (sc-2027; Santa Cruz); CycT1 (sc-899X, Santa Cruz); Cdk9 (sc-8338X, Santa Cruz Biotechnologies); H3K4me1 (ab8895, Abcam); H3K4me3 (ab8580, Abcam); H3K9me3 (ab8898, Abcam); H3K27me3 (ab6002, Abcam); H3K36me3 (ab9050, Abcam); H3K79me3 (ab2621, Abcam); H3K36me3 (ab4729, Abcam); H3 (ab1971, Abcam) Dot1L (A300-954A, Bethyl); SetD2 (ab69836, Abcam); MED1 (A300-793A, Bethyl); TBP (ab28450, Abcam); p300 (sc-585X, Santa Cruz Biotechnologies); FLAG M2 (F1804, Sigma); Tat (ab43014; Abcam). qRT-PCR was performed on a 7500 ABI Fast Real-Time PCR System (Applied Biosystems) as indicated above using the following primers (forward, reverse) with the sign (-/+) and number indicating the position respective to the TSS:

CD69 –17913 (TTGTTGGCCTGAAGTTTTCC, ATACGGATTCACAGCCGAAC)
CD69 –3955 (AATCCAGGGTGAGACGTCAG, GGAAGTCTGTGGTCCCTGTT)
CD69 –63 (AATCCCACTTTCCTCCTGCT, GCCGCCTACTTGCTTGACTA)
CD69 + 2046 (GGTAAACTGGACCAAGAGAAGTTGCC, AATCTGAGTGCCGATCTGTGATGG)

CD69 + 5897 (ACTGTGAAGAGGAGCTGG, GTGTTCCTCTCTACCTGCGTATCG)
CD1E −1499 (GTCCTTGTGATAGTTTGCTGAGAAT, CAAATGTCCAACAATGATAGACTG)
CD1E + 4 (CACAAGAGCAGGGAGAAAATCTGGA, GACAGACAAACCTCTCCCTCTG)
CD1E + 1341 (GGGCCCAGAACATCTGTAAAG, AGAGCAGGTGCTGCATCTTT)
CD1E + 5215 (GGCCAAAGCTCAGAAGAGTGCAG, AGAGCAGGTGCTGCATCTTT)
ADCYAP1 −8465 (TCTCCCCTTCCTGATGATTG, CCCAGAGCACACAGCTAACA)
ADCYAP1 + 1 (AGACACCAACGCCAGACG, GGAGAGCTGCCAGGTAGGAC)
ADCYAP1 + 3713 (TTCCAGGGAGGTTTTGCTATAA, CTGTTTGGGTCCATAAGTGTCC)
ADCYAP1 + 6242 (AGCTCCAACAGACCCTGAGA, ATCTGATTGCTGGGTGAAGG)
RAG1 −3544 (GCCAGTTCATGGATTCAACAAC, TGCGGGCTTTCTGATTTTAGCTAC)
RAG1 + 8 (TGAGAAACAAGAGGGCAAGGAGAGA, CCTGGCCAAAGTGTTGAGATGTCTG)
RAG1 + 5761 (AAGGTTTTCCGGATCGATGTG, GGGCACTGCTAAACTTCCTGTGCAT)
RAG1 + 12,971 (TCCAAGATGCAATGGTGGTA, GTGGGACATGGTGTCAACAG)
PPM1H + 3755 (CCAGGCTAACACACCATGAC, TTGTTAATGCATGGGACAGG)
EOMES −5777 (AAGGCACCCTTAACTGGATG, TCTTGCTCTGCACTTGCTCT)
EOMES −57 (GGGCTGTCACTAGCTGCTTT, CAGGCGACTTGATCCAATTA)
EOMES + 3289 (GGGTGTTGTTGTTATTTGCG, TGTATGTTCACCCAGAGTCTCC)
EOMES + 9699 (GGAGACAGGACAGTCGAGGT, CTAACTGTGTAGCGCGGAAA)
FAM46C −182 (AGGTGTCCCACTAACTCCGA, TGGTTGCCAAGAGACGAGTA)
VAV3 −3936 (TCCTACTGCCCTTGAGTCCT, AAAGTGCCCAGAAGAGTGCT)
VAV3 + 91 (GTAGGAAACGCCAAAGTGGT, TCATGCGGCTAATTTCTTTG)
VAV3 + 102770 (ACAGGCACAACAGTCAAAGG, TGGTTAATTCCAGTGGGTGA)
CD82 −10659 (TTAGGCCAGCTGTCTCATTG, AGGGCAGACATGGAATTAGG)
CD82 + 4180 (CCCACCTGAAGCACCTTTAT, GGGCCGAGATCTCAACTCTA)
CD82 + 28,894 (GGATCCTAACTGCCAAGCAT, CAGAGGCCCAGAGAGGTTAG)
HSPA8 −3885 (CTGGTAGTAGAGCCCTTGCC, CGGACAATATGATCTGCCAA)
HSPA8 −103 (CTGCCCTTACAAGACCCAAT, GGTGAGTGCGTTATCGTGAG)
HSPA8 + 3576 (TTAGCTAGACGCCCTTAGGC, TGTGGACAAGAGTACGGGAA)
FBLN2 + 622 (CACACACGCACTCACACAAG, GAGAGGGAGGATGTGGTGAC)

## ChIP-seq library preparation and sequencing

ChIP DNA was quantified on a Qubit® 2.0 Fluorometer (Life Technologies) and ~10 ng DNA were submitted to the McDermott Center Sequencing Core at UT Southwestern Medical Center for library preparation and high-throughput sequencing using the SOLiD™ ChIP-seq Kit with modifications for the 5500xl instrument (Applied Biosystems). Samples were end repaired, 3'-end adenylated and barcoded with multiplex adapters (Applied Biosystems). After purification with Ampure XP beads (Beckman Coulter), samples were PCR amplified (~14 cycles), size selected with Ampure XP beads, and quantified on the Agilent 2100 Bioanalyzer. Emulsion PCR was then performed and beads enriched on the EZ-Bead system. Between 20–30 × $10^6$ 50 nt sequencing reads were generated for all ChIP experiments in the two cell lines. We uniquely mapped between 70–80% reads to the human genome. High-throughput sequencing was analyzed as described in the computational analysis section below.

## Computational analysis

All scripting was performed using python 2.7.6. All ChIP-seq binding events were loaded into a custom MySQL database to allow for efficient comparisons of multiple factors' binding loci.

### Mapping and sorting

The output from the SOLiD ChIP-seq was aligned to the reference human genome (GRCh37/hg19) using Bowtie v1.0.0. (*Langmead et al., 2009*), allowing 1 mismatch (-v 1). Sequences aligning to multiple locations in the reference genome were discarded. -S was specified to deliver.sam output, which was then sorted and converted to binary (.bam) using Samtools (*Li et al., 2009*).

## ChIP-seq peak calling

The sam files for all ChIP-seq marks were analyzed for binding events using peak calling in the MACS2 software (*Zhang et al., 2008*). When determining peaks, duplicate read alignments were discarded to avoid amplification artifacts from the PCR. An FDR threshold of <0.05 was applied for statistical significance. This methodology was used to call FLAG peaks in both the Tat and GFP cell lines, and only FLAG peaks present in the Tat, but not in the GFP, cell line were considered valid Tat binding events. The GFP:SF cell line was used to control for the presence of the FLAG epitope as well as any inherent chromatin biases in the Jurkat cell lines.

## ChIP-seq data visualization

MACS2 was run in the callpeak mode with the –*B* and –*SPMR* flags set to generate signal pileup tracks in bedGraph format on a per million reads basis. This allows for direct comparison between the GFP and Tat cell lines, regardless of any differences in depth of the sequencing experiment. The pileup tracks were visualized and images were generated using Integrative Genomics Viewer (IGV) (*Robinson et al., 2011*). These normalized pileup tracks also served as the input for many data visualizations as noted below.

## Modified traveling ratio algorithm

Rahl et al. utilized a ratio of the levels of promoter-proximal and distal Pol II within a gene as a probe to determine the rate of transcription phase change from initiation to elongation on a per-gene basis (*Rahl et al., 2010*). In our Pol II ChIP-seq dataset, we noted that Pol II density is elevated above active transcription levels not only at the promoter-proximal location, but also at discrete positions within bodies of actively transcribed genes. These positions are often accompanied by H3K27Ac chromatin marks. We found that these positions are three-dimensionally positioned near the TSS and are actually representative of a form of paused Pol II, not actively transcribing Pol II. Therefore, our modified traveling ratio (mTR) is calculated as follows: promoter-proximal Pol II ChIP reads/promoter-distal Pol II ChIP reads. Promoter-proximal Pol II ChIP reads are considered to be all reads aligned between –30 to +1000 bp of the TSS and all reads present in distal locations having a H3K27Ac peak (as determined by the method provided above) within 500 bp and having a density greater than 5× the average Pol II density between TSS+1001 bp to TTS. Distal Pol II reads are considered to be all those from TSS+1001 bp to TTS that do not fit the criteria for promoter-proximal Pol II as indicated above. The mTR is a custom software and is publicly available. This is availble as *Source code 1*.

## Cis-regulatory motif discovery

To determine conserved DNA motifs associated with Tat, 200 bp fragments of DNA were selected, centered on the summits of Tat peaks found by MACS2 (*Zhang et al., 2008*). These fragments were submitted to the MEME suite (*Bailey et al., 2009*) and motifs were considered significant when having an e-value less than 0.05.

## Generation of Marban et al. pileup track

*Marban et al., (2011*) did not publish processed visualization data such as bigwig or bedGraph pileup files. Therefore, to visually compare the fidelity of our Tat ChIP-seq data with the data generated by Marban et al., we retrieved Marban's raw ChIP-seq read data published in GEO (GSE30738) and processed it using the pipeline described above for mapping and peak calling/pileup track generation. Notably, using the strict p-value cutoff we chose for peak calling with our dataset, MACS2 did not return any peaks when processing the Marban data.

## Annotation of ChIP-seq peaks to genes

HOMER (*Heinz et al., 2010*) annotatePeaks.pl was used to relate ChIP-seq peak location information to the corresponding genes in which those peaks are found. Annotations were made based on the GRCh37/hg19 annotation reference.

## Generation of ChIP-seq heatmaps

We used HOMER (*Heinz et al., 2010*) annotatePeaks.pl to generate density matrices using normalized bedGraph pileup tracks of ChIP-seq marks as input. The *–hist* flag was set with a parameter of 50 bins and the *–ghist* flag was also set to generate a density matrix. Certain marks' heatmaps were generated using a fixed length window (i.e. *–size 6000*) while others were generated based on the entire transcribed region of a set of genes that differ in length (*-size given* with a peakfile containing a bed-style list of TSS and TTS coordinates for each gene in the set).

## Generation of ChIP-seq metagene plots

HOMER (*Heinz et al., 2010*) annotatePeaks.pl was used to generate metagene plots based on the average profile of a series of ChIP-seq marks in normalized bedGraph pileup tracks. The *–hist* flag was set using 50 bins for all metagene profiles. For certain marks, we were only interested in a fixed-length window, so the *–size* flag was set to a fixed number of base pairs, that is. H3K4me3 was plotted centered on TSS and extending -/+ 3 kb (*-size 6000*). Other markers show the profile of the entire transcribed regions of a set of genes that differ in length (*-size given* with a peakfile containing a bed style list of TSS and TTS coordinates for each gene in the set). In these cases, the profile is generated based on relative positions within the genes rather than fixed windows (TSS + 2%, TSS + 4%, etc). In these profiles, we also generated a leading and trailing profile of a fixed 2 kb length.

## Generation of box plots

HOMER (*Heinz et al., 2010*) annotatePeaks.pl was used to generate average density figures for particular ChIP-seq marks using the respective normalized bedGraph pileup tracks as input. This is accomplished by passing the *–size* flag alone in addition to the required parameters. In the case of fixed windows, an integer parameter was passed along with the *–size* flag (*-size 6000*), and in the case of whole genes, the *-size given* option is used, indicating that the peakfile contained coordinates of the TSS and TTS for a set of genes. The resulting output contained a set of average densities for the windows specified and these data were plotted in box plot form using GraphPad Prism.

## RNA-seq data analysis

The 50 nt RNA-seq outputs reads from the sequencer Fastq files were aligned to the UCSC hg19 genome using Tophat v2.0.10 overlaid on Bowtie 1.0.0. Cufflinks v1.3.0 was used to assemble exons discovered by Tophat into competent mRNA transcripts associated with the hg19 annotation (*Trapnell et al., 2013*). The GFP cell line was used as a baseline transcriptome to which to compare that of the Tat cell line. The Tat transcriptome assembled by Cufflinks was compared to the GFP transcriptome using Cuffdiff v1.3.0 to generate a list of genes that were significantly differentially expressed, using a q-value cutoff of 0.05. About $30 \times 10^6$ 50 nt sequencing reads were generated for the GFP and Tat samples. For the GFP cell line we uniquely mapped 67.82% and 69.95% of reads (values for the GFP1 and GFP2 duplicate samples) to the human genome. For the Tat cell line we uniquely mapped 72.46% and 77.35% of reads (values for the Tat1 and Tat2 duplicate samples) to the human genome. FPKM per genes were then produced by counting the number of unique mapped reads that fell within exonic regions for all the genes annotated in the Ensemble hg19. The total length of exonic regions was used to normalize read counts per gene.

## Unbiased identification of TAR-like motifs in the human genome

To identify TAR-like signatures we first extracted genomic sequences of the TSG and TDG including up- and down-stream regions of 10 Kb. In a first-pass filtering approach using seedtop (*Shiryev et al., 2007*) from the NCBI BLAST package, we searched for the following pattern: $X_{(2)}$-G-A-T-$X_{(1,2)}$-G-A-$X_{(4,40)}$-T-C-T-C-$X_{(2)}$, yielding 51,559 sequence signatures. After subjecting these sequences to RNA Fold from the Vienna RNA package (*Lorenz et al., 2011*) and calculating the partition function and base-pairing probability matrix, we used the maximum expected accuracy (MEA) structure and isolated sequences that fold in the corresponding bulged hairpin. In detail, we required the following conserved secondary structure (in dot-bracket notation, with x denoting either dot or bracket): $X_{(2)}$-G-A-T-$X_{(1,2)}$-G-A-$X_{(4,40)}$-T-C-T-C-$X_{(2)}$ and $X_{(2)-(-(-.-.(1,2)}$-$(-(-X_{(4,40)})$-$)-)-)-)$-$X_{(2)}$, resulting in 745 signatures covering 225 genes (*Table 3*).

**Table 3.** Statistics of the TAR signature selection.

|  | TSG | TDG | Total |
|---|---|---|---|
| Targets | 247 | 191 | 438 |
| Genomic sequences (with duplicates) | 1089 | 797 | 1886 |
| Isolated target sequences (with duplicates) | 34739 | 16820 | 51559 |
| With correct secondary structure | 493 | 253 | 746 |
| Corresponding genes | 146 | 86 | 232 |

## Response network

A response network was constructed based on published protein interaction and gene-regulatory data using RNA-seq FPKM values as well as the 438 TSG/TDG targets as seed nodes as previously described (*Mata et al., 2011*). Quantitative expression data was superimposed with network information. The response network was inferred from the large network by calculating k-shortest weighed paths between seed-node pairs. A detailed description of the algorithm can be found here (*Cabusora et al., 2005*).

## Gene-set enrichment analysis

Simple, non-weighted gene set enrichment analysis has been performed by using a variety of different gene-set databases including GO and GSEA/MSigDB (*Ashburner et al., 2000*; *Subramanian et al., 2005*). As a standard procedure, we employed the Fisher exact test and used Bonferroni correction for multiple testing. We also performed quantitative enrichment analysis using the GSEA package together with MSigDB (*Subramanian et al., 2005*).

## Acknowledgements

We are grateful to Vanessa Schmid and the McDermott Center at UT Southwestern for the library preparation and high-throughput sequencing, and the Texas Advanced Computing Center (TACC) for data storage and use of computing resources. We thank Amparo Estrada, Kelly Vollbracht, Jennifer McCann and Blake Richardson for preliminary work related to qRT-PCR assays and cloning; Leonardo Estrada for advice in flow cytometry; Lee Kraus and Ralf Kittler for discussions about ChIP-seq and dataset analysis; and Ana Beatriz Silva and Vicente Planelles for advice on protocols to obtain central memory T cells. Research reported in this publication was supported by the National Institute of Allergy and Infectious Diseases of the NIH under Award Numbers R00AI083087, R56AI106514, and R01AI114362, and The Welch Foundation grant I-1782 to ID; and NIH training grant T32 2T32AI007520-16 to RPM. JER was funded through the Green Fellows program at The University of Texas at Dallas and the SURF Program at UT Southwestern.

## Additional information

### Funding

| Funder | Grant reference number | Author |
|---|---|---|
| National Institute of Allergy and Infectious Diseases | R00AI083087 | Iván D'Orso |
| Welch Foundation | I-1782 | Iván D'Orso |
| National Institute of Allergy and Infectious Diseases | R56AI106514 | Iván D'Orso |
| National Institute of General Medical Sciences | T32 2T32AI007520-16 | Ryan P McNamara |
| National Institute of Allergy and Infectious Diseases | R01AI114362 | Iván D'Orso |

The funders had no role in study design, data collection and interpretation, or the decision to submit the work for publication.

## Author contributions

JER, Conception and design, Acquisition of data, Analysis and interpretation of data, Drafting or revising the article; YTK, Conception and design, Acquisition of data, Analysis and interpretation of data; RPMcN, CVF, Conception and design, Acquisition of data

## Author ORCIDs

Iván D'Orso, http://orcid.org/0000-0002-1409-2351

## Additional files

### Supplementary files

• Supplementary file 1. Enrichment of DE genes in publicly available HIV infection datasets and functional annotation of canonical pathways at the direct Tat target genes.

• Source code 1. Code used to calculate mTR. The mTR algorithm accepts bedgraph files as input and calculates cumulative promoter-proximal paused Pol II occupancy relative to cumulative Pol II occupancy in the gene body after identifying and removing intragenically paused Pol II at sites of high H3K27Ac levels.

### Major datasets

The following datasets were generated:

| Author(s) | Year | Dataset title | Dataset URL | Database, license, and accessibility information |
|---|---|---|---|---|
| Jonathan E Reeder, Ivan D'Orso | 2015 | Tat controls RNA Polymerase II and the epigenetic landscape to precisely rewire cellular transcriptional programs | http://www.ncbi.nlm.nih.gov/geo/query/acc.cgi?acc=GSE65689 | Publicly available at the Gene Expression Omnibus (accession no. GSE65689). |
| | 2011 | Genome-wide binding map of the HIV Tat protein to the human genome | http://www.ncbi.nlm.nih.gov/geo/query/acc.cgi?acc=GSE30738 | Publicly available at the Gene Expression Omnibus (accession no. GSE30738). |

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
