## [Decision Letter]

Thank you for submitting your work entitled "HIV Tat controls RNA Polymerase II and the epigenetic landscape to transcriptionally reprogram immune target cells" for peer review at *eLife*. Your submission has been overall favorably evaluated by James Manley (Senior Editor), Michael Green (Reviewing Editor), and two other expert reviewers, one of whom is Matjaz Barboric.

The reviewers have discussed the reviews with one another and the Reviewing Editor has drafted this decision to help you prepare a revised submission.

The manuscript by Reeder et al. studies the role of the HIV-1 Tat protein on expression of cellular genes. Using a series of genome-wide approaches they show that Tat can both stimulate or repress transcription of cellular genes, and can act at the level of initiation or elongation. They show that Tat promotes initiation through increased recruitment of RNA polymerase II. They also show that Tat is recruited to cellular genes by multiple mechanisms including through interaction with DNA-bound ETS1.

This manuscript arguably represents the most thorough characterization of the role of Tat on cellular gene expression, which has been suggested to be important for viral replication. In general, the authors' conclusions are supported by the presented data. However, the three reviewers raised a number of specific concerns that need to be addressed in a revised manuscript:

1) The authors have used an artificial Tat-inducible Jurkat cell line for their experiments. They present a convincing argument why this artificial system has facilitated the genome-wide analyses. Nonetheless, because the authors believe their results are relevant to normal HIV infection of primary cells, there needs to be some experimental evidence to support this conclusion. There is no reason the authors can't validate some of their key results on specific genes (e.g. effect of Tat on initiation or elongation, Tat occupancy, effect of ETS1 knockdown) in primary T cells or macrophages infected by HIV-1.

2) Figure 1. The authors' use of the minimal system to define direct cellular genes regulated by Tat is reasonable. However, it is less clear (a) whether the predicted GO biological processes of direct Tat targets are retained in the context of viral infection and (b) how would the predicted processes promote viral replication. To address (a), could the authors use publically available data sets on DEGs in HIV infected cells to strengthen their argument on the role of Tat in the transcriptional reprogramming of target cell for the viral benefit? Also, the authors should perform additional RT-qPCRs (rather than for 1 gene from each category) from the material of infected Jurkat cells (Figure 1—figure supplement 5). For (b), it would be beneficial to the non-specialists had the authors described better how the predicted GO biological processes might promote viral replication.

3) Figure 2–Figure 7. Given the central role of P-TEFb in the stimulation of HIV-1 transcription elongation by Tat and the authors' observation that Tat activates and represses transcription elongation at class II TSGs and TDGs, respectively, the authors should perform ChIP-qPCR assay at select genes to provide evidence whether Tat promotes or precludes the recruitment of P-TEFb to the class II gene categories. The authors should also include Ser2 phosphorylation status of Pol II at these genes.

4) Figure 9. Following the ETS1-Tat interaction data (coimmunoprecipitation assays, ChIPs assays in CTRL and ETS1 knockdown cells), the authors propose a model that ETS1 recruits Tat to a subset of cellular genes (such as CD69) to reprogram cellular transcription. First, given that the mutant C22A Tat protein fails to occupy the CD69 promoter (Figure 1—figure supplement 3), the authors should examine if this mutant protein fails to bind ETS1. Importantly, the authors should provide evidence that ETS1 is critical for transcriptional activation of CD69 by Tat as well as for Tat-mediated increase in the levels of Pol II and the chromatin marks coinciding with increased transcription initiation and elongation at this class I TSG.

5) The manuscript would benefit from re-writing to better put what was known about Tat in context and to more accurately frame our state of knowledge concerning chromatin and transcription mechanisms. There are a number of statements that could be misleading to the field. Examples are:

"Histone modifications […] allow for precise demarcation of the position and activity of gene promoters, coding units and enhancers" (Introduction). Promoter and enhancer regions as defined by histone modifications are far from precise, and several modifications do not correlate with the level of gene activity- as shown for H3K4me3 in this article.

"The majority of differentially expressed genes do not experience transcription initiation changes…" (in the subsection “Global analysis of chromatin signatures indicates that Tat stimulates transcription at both the initiation and elongation steps”). For a gene to be up-regulated, there must be an increase in transcription initiation. You can't make more RNA if more polymerases aren't loaded onto the promoter. It is totally true that not every initiated Pol II will make a mature transcript, but it is not true that you can make a mature transcript without loading Pol II.

Most importantly, the authors do not explain fully the rich literature on the role of Tat in pause release that has been published previously (over decades). Much of what is shown herein is not surprising, and the proper context would be helpful.

6) It is not clear what is learned by the analysis of histone marks +/- Tat. The findings are completely in line with previous publications showing that the level of histone marks such as H3K4me3 and H3K79me3 generally correspond to transcription activity, but aren't very good at indicating actual transcription levels. But I don't see what this tells us about Tat activity. The authors should acknowledge that the histone modification data is likely to represent indirect effects of changes in transcription rather than direct targets of Tat activity.

7) The number of genes in 'Class I TSGs' is so small and the signals at these genes so low as to be very difficult to distinguish from noise. It appears that there are fewer than 20 genes in Class I TSGs (Figure 2), which may represent the edge of the distribution in class II, not a distinct class of their own. The threshold to separate these out was defined completely arbitrarily. The authors should very clearly note in each figure the total number of genes they put in each class, and to refrain from drawing conclusions based on very small gene sets.

8) Figure 7. By looking at Figure 7—figure supplement 1, the combined number of class I and class II TDGs does not seem to be close to the total number of TDGs (212). Could the authors state in the text how many TDGs are of class I and class II? If there are many TDGs that do not fall into the class I & II categories, could the authors speculate on how might these genes be repressed by Tat?

---

## [Author Response]

*1) The authors have used an artificial Tat-inducible Jurkat cell line for their experiments. They present a convincing argument why this artificial system has facilitated the genome-wide analyses. Nonetheless, because the authors believe their results are relevant to normal HIV infection of primary cells, there needs to be some experimental evidence to support this conclusion. There is no reason the authors can't validate some of their key results on specific genes (e.g. effect of Tat on initiation or elongation, Tat occupancy, effect of ETS1 knockdown) in primary T cells or macrophages infected by HIV-1.*

We agree with this point. We have now incorporated a completely new experiment in primary CD4+ T cells infected with replication-competent HIV-1. We decided to use primary CD4+ T cells (and not macrophages) because they are more closely related to the Jurkat CD4+ T-cell line used in our minimalistic study. We now show that specific TDG and TSG selected from our screen are in fact deregulated during a time course infection experiment in primary T cells. The results are shown in the new Figure 1—figure supplement 8, the data discussed in the text (last paragraph of the subsection “Genomic domains occupied and regulated by Tat in CD4+ T cells”), and the materials and methods included (in the subsection “Primary CD4+ T cell isolation, stimulation and HIV infection”). We also have demonstrated that the class I and class II TSG/TDG identified in the minimalistic Jurkat system (at least for the target genes examined by qRT-PCR) are also regulated at the initiation and elongation steps, respectively, during HIV infection of primary T cells. The results are shown in the new Figure 1—figure supplement 8 and the data discussed in the text (in the subsection “Global analysis of chromatin signatures indicates that Tat stimulates transcription at both the initiation and elongation steps”, eighth paragraph for TSG and in the subsection “Global analysis of chromatin signatures and Pol II distribution indicate that Tat downregulates transcription by blocking both initiation and elongation”, end of first paragraph for TDG). Despite these significant additions to the revised manuscript, we were unable to perform the ETS1 knockdown in primary cells to measure Tat effects during infection. There are several explanations for the lack of success. First, it is extremely difficult to perform RNAi in primary cells due to the presence of several endogenous barriers. Second, also importantly, ETS1 knockdown affects cell growth and viability and thus compromises the infection step. Also, unfortunately, we were not able to perform ChIP assays for Pol II and other markers because of the large number of cells required for these experiments. Nonetheless, we have provided compelling evidence that the Tat target genes identified using our genome-wide analysis in Jurkat cells are relevant to normal HIV-1 infection of primary CD4+ T cells and we have provided evidence of Tat-mediated transcriptional control at the initiation and elongation steps. We hope the reviewers value the investment made and the data yielded from this new set of experiments in primary cells.

*2) Figure 1. The authors' use of the minimal system to define direct cellular genes regulated by Tat is reasonable. However, it is less clear (a) whether the predicted GO biological processes of direct Tat targets are retained in the context of viral infection and (b) how would the predicted processes promote viral replication. To address (a), could the authors use publically available data sets on DEGs in HIV infected cells to strengthen their argument on the role of Tat in the transcriptional reprogramming of target cell for the viral benefit? Also, the authors should perform additional RT-qPCRs (rather than for 1 gene from each category) from the material of infected Jurkat cells (Figure 1—figure supplement 5). For (b), it would be beneficial to the non-specialists had the authors described better how the predicted GO biological processes might promote viral replication.*

We agree with this point. We have performed further analysis as the reviewers suggested and answered the two points as indicated below:

(a) We used gene expression datasets from 13 publications studying diverse aspects on HIV infection and replication. From these 13 publications we extracted 48 gene sets that were used together with 14 HIV relevant datasets from the Molecular Signature Database (MSigDB) of the Broad Institute. After enrichment analysis we could validate that TSG as well as TDG significantly enrich some of these 62 gene sets, indicating their role in viral infection. The new data is included in the new Figure 1—figure supplement 10, new [Supplementary-material SD1-data], and discussed in the text (in the subsection “The direct Tat target genes share common functional annotations and are enriched in pathways beneficial for the virus”).

(b) We added a paragraph to better relate the identified GO biological processes to viral replication. In addition, we added results of a functional enrichment analysis using the canonical pathway dataset (referred to as c2.cp) of the MSigDB that provides a more fine-grained functional annotation. We now were able to link high-level GO processes with MSigDB pathways and their corresponding sub-network or clusters in our response network. By this means, we identified 27 TSG and 45 TDG that enrich the MSigDB pathways (new Figure 1—figure supplement 10) and key-processes that promote viral replication (such as the T-cell signaling and co-stimulatory pathways, among others), in agreement with our original network analysis (Figure 1—figure supplement 9). The new data is included in the new Figure 1—figure supplement 10, new [Supplementary-material SD1-data] (for TSG) and new [Supplementary-material SD1-data] (for TDG) and discussed in the text (in the subsection “The direct Tat target genes share common functional annotations and are enriched in pathways beneficial for the virus”).

*3) Figure 2–Figure 7. Given the central role of P-TEFb in the stimulation of HIV-1 transcription elongation by Tat and the authors' observation that Tat activates and represses transcription elongation at class II TSGs and TDGs, respectively, the authors should perform ChIP-qPCR assay at select genes to provide evidence whether Tat promotes or precludes the recruitment of P-TEFb to the class II gene categories. The authors should also include Ser2 phosphorylation status of Pol II at these genes.*

We agree with this point. We have performed the corresponding ChIP-qPCR at selected (two) class II TSG and TDG. The results are shown in the new Figure 3—figure supplement 2 and the data discussed in the text (in the subsection “Tat promotes Pol II recruitment and pause release to induce initiation and elongation at different gene classes”, last paragraph for TSG and in the subsection “Global analysis of chromatin signatures and Pol II distribution indicate that Tat downregulates transcription by blocking both initiation and elongation”, second paragraph for TDG). Importantly, the data revealed that Tat increases levels of P-TEFb and Ser2P-CTD at class II TSG, while decreasing them at class II TDG, consistent with the fluctuations in transcript levels and the changes in histone modifications at promoters and gene bodies.

*4) Figure 9. Following the ETS1-Tat interaction data (coimmunoprecipitation assays, ChIPs assays in CTRL and ETS1 knockdown cells), the authors propose a model that ETS1 recruits Tat to a subset of cellular genes (such as CD69) to reprogram cellular transcription. First, given that the mutant C22A Tat protein fails to occupy the CD69 promoter (Figure 1—figure supplement 3), the authors should examine if this mutant protein fails to bind ETS1. Importantly, the authors should provide evidence that ETS1 is critical for transcriptional activation of CD69 by Tat as well as for Tat-mediated increase in the levels of Pol II and the chromatin marks coinciding with increased transcription initiation and elongation at this class I TSG.*

We agree with this point and we have now incorporated the corresponding co-IP data showing that wild-type Tat but not the non-functional mutant (C22A) binds ETS1. The new results are shown in the new Figure 9—figure supplement 1 and the data is mentioned in the text (in the subsection “ETS1 recruits Tat to chromatin to reprogram cellular transcription” first paragraph). We also agree with the second point. We have now provided evidence that ETS1 is critical for Tat-mediated transcriptional activation of *CD69*, and three other target genes (new Figure 9—figure supplement 1 and data discussed in the third paragraph of the aforementioned subsection) as well as for the increase in the levels of Pol II and the chromatin marks (H3K4me3 and H3K79me3) coinciding with increased transcription initiation and elongation, respectively, at class I TSG (Figure 2 and new Figure 2—figure supplement 3).

*5) The manuscript would benefit from re-writing to better put what was known about Tat in context and to more accurately frame our state of knowledge concerning chromatin and transcription mechanisms. There are a number of statements that could be misleading to the field. Examples are:*

"Histone modifications […] allow for precise demarcation of the position and activity of gene promoters, coding units and enhancers" (Introduction). Promoter and enhancer regions as defined by histone modifications are far from precise, and several modifications do not correlate with the level of gene activity- as shown for H3K4me3 in this article.

"The majority of differentially expressed genes do not experience transcription initiation changes…" (in the subsection “Global analysis of chromatin signatures indicates that Tat stimulates transcription at both the initiation and elongation steps”). For a gene to be up-regulated, there must be an increase in transcription initiation. You can't make more RNA if more polymerases aren't loaded onto the promoter. It is totally true that not every initiated Pol II will make a mature transcript, but it is not true that you can make a mature transcript without loading Pol II.

*Most importantly, the authors do not explain fully the rich literature on the role of Tat in pause release that has been published previously (over decades). Much of what is shown herein is not surprising, and the proper context would be helpful.*

We agree with these three points and we apologize for the misleading statements. We have rewritten parts of the manuscript and included new sections to better discuss what was known about Tat in context and to more accurately frame our state of knowledge concerning chromatin and transcription mechanisms. Examples are in the Introduction and in the subsection “Global analysis of chromatin signatures indicates that Tat stimulates transcription at both the initiation and elongation steps”, among others.

*6) It is not clear what is learned by the analysis of histone marks +/- Tat. The findings are completely in line with previous publications showing that the level of histone marks such as H3K4me3 and H3K79me3 generally correspond to transcription activity, but aren't very good at indicating actual transcription levels. But I don't see what this tells us about Tat activity. The authors should acknowledge that the histone modification data is likely to represent indirect effects of changes in transcription rather than direct targets of Tat activity.*

We agree with the reviewers that it is possible that the epigenetic landscape at Tat target genes may be a result of indirect rather than direct Tat activity. However, it is worth noting the protein-protein interactions detected between Tat and the histone modifying enzymes Dot1L and SetD2 (in the subsection “Global analysis of chromatin signatures indicates that Tat stimulates transcription at both the initiation and elongation steps”, and Figure 2—figure supplement 5). This potentially indicates more than an indirect role in chromatin remodeling. In either case, direct or indirect, we found it worth noting that different groups of TSG and TDG exhibit different changes to the chromatin modification landscape, suggesting different modes of both activation and repression. Also, our data regarding a genome-wide histone landscape helps inform not only the specifics of Tat action, but also provides general insights regarding the role of histone modification at promoters and enhancers (some along the lines of previous investigations), including:

Identification of correct isoforms or selection of internal non-canonical, intragenic TSS,

Presence of H3K79me3 patterns surrounding internally/intragenically paused Pol II,

Potential role for H3K79me3 mark in transcription repression (not only activation).

Furthermore, per the reviewers’ suggestion, we have acknowledged in the text that histone modifications may represent (in some cases) indirect effects of changes in transcription and not a direct consequence of Tat function on the genome (Discussion).

*7) The number of genes in 'Class I TSGs' is so small and the signals at these genes so low as to be very difficult to distinguish from noise. It appears that there are fewer than 20 genes in Class I TSGs (Figure 2), which may represent the edge of the distribution in class II, not a distinct class of their own. The threshold to separate these out was defined completely arbitrarily. The authors should very clearly note in each figure the total number of genes they put in each class, and to refrain from drawing conclusions based on very small gene sets.*

The class I TSG shows evidence of very different modes of stimulation than the class II TSG, rather than different degrees of the same modes of stimulation. Our class I thresholds were selected to isolate genes that appear to be transcriptionally inactive prior to Tat stimulation, and then are activated at the level of transcription initiation. The threshold for class II TSG was selected to identify genes that are already transcriptionally active prior to Tat and are then further stimulated at the level of elongation. Our findings show that a very small number of genes are activated from an “off” state (17 class I TSG), which, as the reviewers correctly note, does not allow for broad conclusions about one generic method of Tat stimulation. It does, however, demonstrate that there is more than one mechanism utilized by Tat to activate/stimulate target genes, and that different genes are targeted in different manners. With regards to the signal at the class I genes being indistinguishable from noise, the ChIP-seq in the control cell line (GFP) provides compelling evidence that this is not the case. Our levels of signal to input were thoroughly vetted for reliability. Assuming that the reviewers are speaking about a marker such as H3K4me3, the levels of normalized tag densities do appear low, but the differences between the marker in the Tat vs. GFP cell lines are remarkable. The normalization to tag density per million reads yields small numbers in the metagene plots, but these nominal scales are far less important than the difference between control and test. Further, because the class I genes are nearly transcriptionally silent prior to perturbation by Tat, it does not require much of a change in the initiation markers to yield a noticeable increase in expression. We have now included examples of raw data (Genome browser views in IGV) in the modified Figure 2—figure supplement 1 so that the reader can see examples of real plots and nominal densities prior to averaging, in addition to the previous H3K4me3 levels in a gene-by-gene basis (modified Figure 2—figure supplement 1).

*8) Figure 7. By looking at Figure 7—figure supplement 1, the combined number of class I and class II TDGs does not seem to be close to the total number of TDGs (212). Could the authors state in the text how many TDGs are of class I and class II? If there are many TDGs that do not fall into the class I & II categories, could the authors speculate on how might these genes be repressed by Tat?*

The reviewers are correct about indicating that “the combined number of class I and class II TDG does not seem to be close to the total number of TDG”. We apologize for the oversight of not including this information. We have now incorporated the number of class I and II TDG in every experiment (text and figure legends). We have explained in the original submission the approach used to classify TDG as class I or II. The reason for not including all TDG was that most of the genes might be a blend of class I and class II (probably regulated at both the initiation and elongation levels), or alternatively, regulated by unknown mechanisms. To make our findings of multiple modes of Tat action clear, we highlighted the most demonstrative cases of each mode and avoided attempting to tease out the mixed-mode genes. Those examples provide the best evidence that Tat is working on multiple transcription steps. We consider that defining all the mechanisms would be interesting and informative, but it may take additional data collection. For example, RNA-seq is not a direct measure of transcription and to strengthen the TDG classification we may need to collect transcriptional data by GRO-seq or NET-seq (Discussion). This is well beyond the scope of one single manuscript.